# QVGen: Pushing the Limit of Quantized Video Generative Models

**Yushi Huang**[1,3*]  **Ruihao Gong**[2,3†]  **Jing Liu**[4]  **Yifu Ding**[2]  **Chengtao Lv**[3,5*]
**Haotong Qin**[6]  **Jun Zhang**[1†]
[1]Hong Kong University of Science and Technology   [2]Beihang University   [3]SenseTime Research
[4]Monash University   [5]Nanyang Technological University   [6]ETH Zürich
{huangyushi1, gongruihao, lvchengtao}@sensetime.com  liujing_95@outlook.com
yifuding@buaa.edu.cn  haotong.qin@pbl.ee.ethz.ch  eejzhang@ust.hk

## Abstract

Video diffusion models (DMs) have enabled high-quality video synthesis. Yet, their substantial computational and memory demands pose serious challenges to real-world deployment, even on high-end GPUs. As a commonly adopted solution, quantization has achieved notable successes in reducing cost for image DMs, while its direct application to video DMs remains ineffective. In this paper, we present *QVGen*, a novel quantization-aware training (QAT) framework tailored for high-performance and inference-efficient video DMs under extremely low-bit quantization (*i.e.*, 4-bit or below). We begin with a theoretical analysis demonstrating that reducing the gradient norm is essential to facilitate convergence for QAT. To this end, we introduce auxiliary modules ($\Phi$) to mitigate large quantization errors, leading to significantly enhanced convergence. To eliminate the inference overhead of $\Phi$, we propose a *rank-decay* strategy that progressively eliminates $\Phi$. Specifically, we repeatedly employ singular value decomposition (SVD) and a proposed rank-based regularization $\gamma$ to identify and decay low-contributing components. This strategy retains performance while zeroing out additional inference overhead. Extensive experiments across 4 state-of-the-art (SOTA) video DMs, with parameter sizes ranging from 1.3B$\sim$14B, show that QVGen is *the first* to reach full-precision comparable quality under 4-bit settings. Moreover, it significantly outperforms existing methods. For instance, our 3-bit CogVideoX-2B achieves improvements of $+25.28$ in Dynamic Degree and $+8.43$ in Scene Consistency on VBench. Code and models are available at https://github.com/ModelTC/QVGen.

## 1 Introduction

Recently, advancements in artificial intelligence-generated content (AIGC) have led to significant breakthroughs in text (Touvron et al., 2023; DeepSeek-AI et al., 2025), image (Xie et al., 2025; Labs, 2024), and video synthesis (WanTeam et al., 2025; Kong et al., 2025). The development of video generative models, driven by the powerful diffusion transformer (DiT) (Peebles & Xie, 2023) architecture, has been particularly notable. Leading video diffusion models (DMs), such as closed-source OpenAI Sora (OpenAI, 2024) and Kling (Kuaishou, 2024), and open-source `Wan` (WanTeam et al., 2025) and CogVideoX (Yang et al., 2025), can successfully model *motion dynamics*, *semantic scenes*, *etc.* Despite their impressive performance, these models demand high computational resources and substantial peak memory, especially when generating long videos at a high resolution. For example, `Wan` 14B requires more than 30 minutes and 50GB of GPU memory to generate a 10-second $720p$ resolution video clip on a single H100 GPU. Even worse, deploying such models is infeasible on most customer-grade PCs, let alone resource-constrained edge devices. As a result, their practical applications across various platforms face considerable challenges.

In light of these problems, model quantization, which maps high-precision (*e.g.*, `FP16`/`BF16`) data to low-precision (*e.g.*, `INT8`/`INT4`) formats, stands out as a compelling solution. For instance,

---

*Work done during internships at SenseTime Research.

†Corresponding authors.

**(a)** `BF16`                    **(b)** `W4A4` QVGen (Ours)

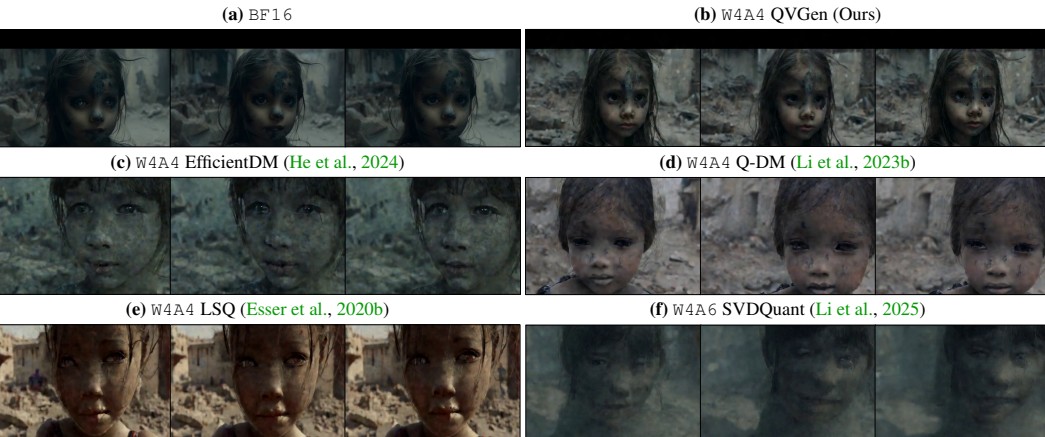

**(c)** `W4A4` EfficientDM (He et al., 2024)          **(d)** `W4A4` Q-DM (Li et al., 2023b)

**(e)** `W4A4` LSQ (Esser et al., 2020b)          **(f)** `W4A6` SVDQuant (Li et al., 2025)

Text prompt: *"In the haunting backdrop of a war-torn city, where ruins and crumbled walls tell a story of devastation, a poignant close-up frames a young girl. Her face is smudged with ash, a silent testament to the chaos around her. Her eyes glistening with a mix of sorrow and resilience, capturing the raw emotion of a world that has lost its innocence to the ravages of conflict."*

**Figure 1:** Comparison of samples generated by CogVideoX-2B (Yang et al., 2025) with a fixed random seed. "`WxAy`" denotes "$x$"-bit *per-channel* weight and "$y$"-bit *per-token* activation quantization. Our approach far outperforms previous PTQ (*i.e.*, (f)) and QAT (*i.e.*, (c)-(e)) methods. To be noted, methods (c)-(f) have achieved noticeable performance for 4-bit image DMs. More visual results can be found in Sec. P.

employing 4-bit models with fast kernel implementation can achieve a significant $3\times$ speedup ratio with about $4\times$ model size reduction compared with floating-point models on NVIDIA RTX4090 GPUs (Li et al., 2025). However, quantizing video DMs is more challenging than quantizing image DMs, and it has not received adequate attention. As shown in Fig. 1, applying prior high-performing approaches to quantize a video DM into ultra-low bits ($\leq$ 4 bits) is ineffective. In contrast to post-training quantization (PTQ), quantization-aware training (QAT) can obtain superior performance through training quantized weights. Nevertheless, it still leads to severe video quality degradation, as demonstrated by Fig. 1 (a) *vs.* (c)-(e). This highlights the need for an improved QAT framework to preserve video DMs' exceptional performance under 4-bit or lower quantization.

In this work, we present a novel QAT framework, termed *QVGen*. It aims to improve the convergence without additional inference costs of low-bit Quantized DMs for Video Generation.

Specifically, we first provide a theoretical analysis showing that minimizing the gradient norm $\|\boldsymbol{g}_t\|_2$ is the key to improving the convergence of QAT for video DMs. Motivated by this finding, we introduce auxiliary modules $\Phi$ for the quantized video DM to mitigate quantization errors. These modules effectively help narrow the discrepancy between the discrete quantized and full-precision models, leading to stable optimization and largely reduced $\|\boldsymbol{g}_t\|_2$. The quantized DM thus achieves better convergence. Our observation also implies that the significant performance drops (Fig. 1) of the existing SOTA QAT method (Li et al., 2023b) may result from its high $\|\boldsymbol{g}_t\|_2$ (Fig. 3).

Moreover, to adopt $\Phi$ for improving QAT while avoiding its substantial inference overhead, we progressively remove $\Phi$ during training. Upon further analysis, we have found that the amount of small singular values in $\mathbf{W}_\Phi$ (the weight of $\Phi$) increases throughout the training process. This indicates that the quantity of low-contributing components in $\mathbf{W}_\Phi$, which are related to small singular values (Zhang et al., 2015; Yang et al., 2020), grows during QAT. As a result, an increasing number of these components can be removed with minimal impact on training. Leveraging this insight, we introduce a *rank-decay* strategy to progressively shrink $\mathbf{W}_\Phi$. To be more specific, singular value decomposition (SVD) is first applied to recognize $\mathbf{W}_\Phi$'s low-impact components. Then, a rank-based regularization $\gamma$ is utilized to gradually decay these components to $\varnothing$. Such processes (*i.e.*, decompose and then decay) are repeated until $\mathbf{W}_\Phi$ is fully eliminated, which also means that $\Phi$ is removed. In terms of results, this strategy incurs minimal performance impact while getting rid of the extra inference overhead.

To summarize, our contributions are as follows:

▶ We introduce a general-purpose QAT paradigm, called QVGen. To our knowledge, this is *the first* QAT method for video generation and achieves effective 3-bit and 4-bit quantization.
▶ To optimize extremely low-bit QAT, we enhance a quantized DM with auxiliary modules ($\Phi$) to reduce the gradient norm. Our theoretical and empirical analysis validates the effectiveness of this

method in improving convergence.

▶ To eliminate the significant inference overhead introduced by $\Phi$, we propose a *rank-decay* strategy that progressively shrinks $\Phi$. It iteratively performs SVD and applies a rank-based regularization $\gamma$ to obtain and decay low-impact components of $\Phi$, respectively. As a result, this method incurs minimal impact on performance.

▶ Extensive experiments across advancing CogVideoX and Wan families demonstrate the SOTA performance of QVGen. Notably, our W4A4 model is *the first* to show full-precision comparable performance. In addition, we apply QVGen to Wan 14B, one of the largest open-source SOTA models, and observe negligible performance drops on VBench-2.0.

## 2 PRELIMINARIES

**Video diffusion modeling.** The video DM (Ho et al., 2022; Zheng et al., 2024) extends image diffusion frameworks (Li et al., 2023b; Song et al., 2021a) into the temporal domain by learning dynamic inter-frame dependencies. Let $\boldsymbol{x}_0 \in \mathbb{R}^{f \times h \times w \times c}$ be a latent video variable, where $f$ denotes the count of video frames, each of size $h \times w$ with $c$ channels. DMs are trained to denoise samples generated by adding random Gaussian noise $\boldsymbol{\epsilon} \sim \mathcal{N}(\mathbf{0}, \mathbf{I})$ to $\boldsymbol{x}_0$:

$$\boldsymbol{x}_\tau = \alpha_\tau \boldsymbol{x}_0 + \sigma_\tau \boldsymbol{\epsilon}, \tag{1}$$

where $\alpha_\tau, \sigma_\tau > 0$ are scalar values that collectively control the signal-to-noise ratio (SNR) according to a given noise schedule (Song et al., 2021b) at timestep $\tau \in [1, \ldots, N]$[1]. One typical training objective (*i.e.*, predict the noise (Ho et al., 2020b)) of a denoiser $\boldsymbol{\epsilon}_\theta$ with parameter $\theta$ can be formulated as follows:

$$\mathcal{L}(\theta) = \mathbb{E}_{\boldsymbol{x}_0, \boldsymbol{\epsilon}, \mathcal{C}, \tau}[\|\boldsymbol{\epsilon} - \boldsymbol{\epsilon}_\theta(\boldsymbol{x}_\tau, \mathcal{C}, \tau)\|_F^2], \tag{2}$$

where $\mathcal{C}$ represents conditional guidance, like texts or images, and $\|\cdot\|_F$ denotes the Frobenius norm. Additionally, v-prediction (Salimans & Ho, 2022) (*i.e.*, predict $\frac{d\boldsymbol{x}_\tau}{d\tau}$) is also a prevailing option (Yang et al., 2025; WanTeam et al., 2025; Kong et al., 2025) as the target. During inference, we can employ $\boldsymbol{\epsilon}_\theta$ with various sampling methods (Lu et al., 2022; Zheng et al., 2023) progressively denoising from a random Gaussian noise $\boldsymbol{x}_N \sim \mathcal{N}(\mathbf{0}, \mathbf{I})$ to a clean video variable. The raw video is obtained by decoding the variable via a video variational auto-encoder (VAE) (Yang et al., 2025).

**Quantization.** The current video DM based on the diffusion transformer (DiT) (Peebles & Xie, 2023) architecture primarily consists of linear layers. Given an input $\mathbf{X} \in \mathbb{R}^{m \times k}$, a full-precision linear layer with weight $\mathbf{W} \in \mathbb{R}^{n \times m}$ and the layer's quantized version can be formulated as:

$$\mathbf{Y} = \mathbf{W}\mathbf{X}, \hat{\mathbf{Y}} = \mathcal{Q}_b(\mathbf{W})\mathcal{Q}_b(\mathbf{X}), \tag{3}$$

where $\mathbf{Y} \in \mathbb{R}^{n \times k}$ and $\hat{\mathbf{Y}} \in \mathbb{R}^{n \times k}$ [2] represent the outputs of the full-precision (*e.g.*, FP16/BF16) and quantized linear layers, respectively. $\mathcal{Q}_b(\cdot)$ denotes the function of $b$-bit quantization. In this paper, we adopt asymmetric uniform quantization. For example, $\mathcal{Q}_b(\mathbf{X})$ can be represented as:

$$\mathcal{Q}_b(\mathbf{X}) = (\text{clip}(\lfloor \tfrac{\mathbf{X}}{s} \rceil + z, 0, 2^b - 1) - z) \times s, \text{ where } s = \tfrac{\max(\mathbf{X}) - \min(\mathbf{X})}{2^b - 1}, z = -\lfloor \tfrac{\min(\mathbf{X})}{s} \rceil. \tag{4}$$

Here, quantization parameters $s$ and $z$ denote the *scaler* and *zero shift*, respectively. $\text{clip}(\cdot, \cdot, \cdot)$ bounds the integer values into $[0, 2^b - 1]$. To ensure the differentiability of the rounding function $\lfloor \cdot \rceil$ for QAT, straight-through estimator (STE) (Bengio et al., 2013) is widely applied as:

$$\frac{\partial \mathcal{Q}_b(\mathbf{X})}{\partial \mathbf{X}} = \mathbb{I}_{0 \leq \lfloor \frac{\mathbf{X}}{s} \rceil + z \leq 2^b - 1}. \tag{5}$$

Similar to existing works (Li et al., 2023b; Zheng et al., 2025b), we employ the full-precision model as the teacher to guide the training of the quantized model in a knowledge distillation-based (KD-based) manner. Therefore, the training loss can be defined as:

$$\mathcal{L} = \mathbb{E}_{\boldsymbol{x}_0, \mathcal{C}, \tau}[\|\hat{\boldsymbol{\epsilon}}_\theta(\boldsymbol{x}_\tau, \mathcal{C}, \tau) - \boldsymbol{\epsilon}_\theta(\boldsymbol{x}_\tau, \mathcal{C}, \tau)\|_F^2]. \tag{6}$$

where $\hat{\boldsymbol{\epsilon}}_\theta$ denotes the quantized denoiser of a video DM.

---

[1] $N$ denotes the maximum timestep.

[2] Here, $n$ denotes the output channel, $m$ signifies the token dimension, and $k$ is the token number. We omit the batch dimension for clarity.

## 3 QVGEN

Considering the substantial video-quality drops (see Fig. 1) observed in existing QAT methods (Li et al., 2023b; He et al., 2024; Esser et al., 2020b), we believe that the quantized video DM suffers from poor convergence. In the following subsections, we propose QVGen (see Fig. 2) to address this issue while maintaining inference efficiency.

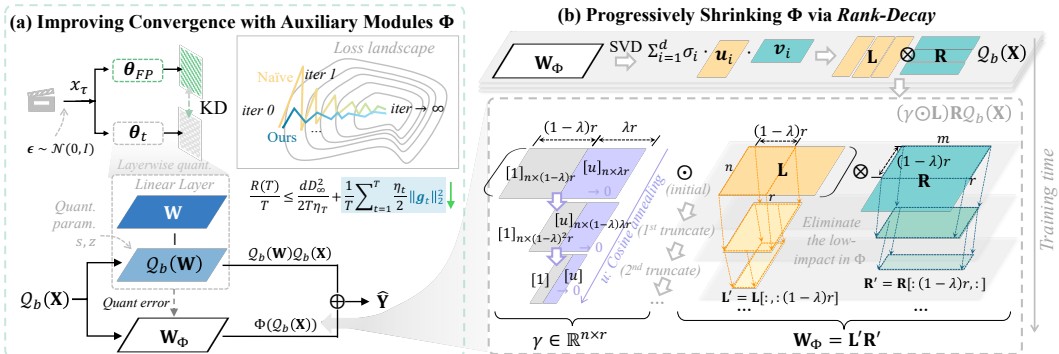

**Figure 2:** Overview of the proposed QVGen. (a) This framework integrates auxiliary modules $\Phi$ to improve training convergence (Sec. 3.1). (b) To maintain performance while eliminating inference overhead induced by $\Phi$, we design a *rank-decay* schedule that progressively shrinks the entire $\Phi$ to $\varnothing$ through *iteratively applying* the following two strategies (Sec. 3.2): (*i*) SVD to identify the low-impact components in $\Phi$; (*ii*) A rank-based regularization $\gamma$ to decay the identified components to $\varnothing$. A detailed procedure can be found in Sec. A.

### 3.1 IMPROVING CONVERGENCE WITH AUXILIARY MODULES $\Phi$

To begin with, we analyze the convergence of a quantized video DM using the regret, which is widely used in analyses of deep learning optimizers (Kingma & Ba, 2017; Luo et al., 2019). It is defined as:

$$R(T) = \sum_{t=1}^{T} f_t(\boldsymbol{\theta}_t) - f_t(\boldsymbol{\theta}^*), \tag{7}$$

where $T$ signifies the total number of training iterations and $f_t(\cdot)$ is the unknown cost function at iteration $t$. Here, $\boldsymbol{\theta}_t$ represents the parameters of the quantized video DM at training step $t$, constrained within a convex compact set $\mathbb{S}^d$, while $\boldsymbol{\theta}^* = \arg\min_{\boldsymbol{\theta} \in \mathbb{S}^d} \sum_{t=1}^{T} f_t(\boldsymbol{\theta})$ is the optimal parameters. In QAT, $\boldsymbol{\theta}_t$ is updated by gradient descent, with the learning rate $\eta_t$ and gradient $\boldsymbol{g}_t$, as:

$$\boldsymbol{\theta}_{t+1} = \boldsymbol{\theta}_t - \eta_t \boldsymbol{g}_t. \tag{8}$$

**Theorem 3.1.** *Assume that $f_t$ is convex[3] and $\forall \boldsymbol{\theta}_i, \boldsymbol{\theta}_j \in \mathbb{S}^d, \|\boldsymbol{\theta}_i - \boldsymbol{\theta}_j\|_\infty \leq D_\infty$. Then the average regret is upper-bounded as:* $\frac{R(T)}{T} \leq \frac{d D_\infty^2}{2 T \eta_T^m} + \frac{1}{T} \sum_{t=1}^{T} \frac{\eta_t^M}{2} \|\boldsymbol{g}_t\|_2^2$[4].

A smaller value of $\frac{R(T)}{T}$ implies a closer convergence to the optimum. Thm. 3.1 (with a proof provided in Sec. B) suggests that for a large $T$ (*i.e.*, $\frac{d D_\infty^2}{2 T \eta_T^m}$ becomes negligible), minimizing $\|\boldsymbol{g}_t\|_2$ is critical for improving convergence behavior of QAT. A lower $\|\boldsymbol{g}_t\|_2$ is typically observed in more stable training processes (Takase et al., 2024; Xie et al., 2024). Therefore, to reduce $\|\boldsymbol{g}_t\|_2$, we aim to stabilize the QAT process by mitigating aggressive training losses (*e.g.*, loss spikes) (Kumar et al., 2025; Li et al., 2024b). Specifically, we introduce a learnable auxiliary module $\Phi$ to enhance each quantized linear layer of a video DM. This trainable module aims to mitigate severe quantization-induced errors during QAT, thereby preventing aggressive training losses. The forward computation of such a $\Phi$-equipped layer becomes:

$$\hat{\mathbf{Y}} = \mathcal{Q}_b(\mathbf{W}) \mathcal{Q}_b(\mathbf{X}) + \Phi(\mathcal{Q}_b(\mathbf{X})), \tag{9}$$

where $\Phi(\mathcal{Q}_b(\mathbf{X})) = \mathbf{W}_\Phi \mathcal{Q}_b(\mathbf{X})$. Here, $\mathbf{W}_\Phi$ is initialized before QAT by the weight quantization error, defined as $\mathbf{W} - \mathcal{Q}_b(\mathbf{W})$. More initialization approaches for $\mathbf{W}_\Phi$ can be found in Sec. L.2.

---

[3]This may not hold for deep networks. Therefore, we also provide a nonconvex convergence analysis in Sec. C.

[4]$\eta_t^M$ and $\eta_t^m$ are the maximum and minimum values of $\eta_t$, respectively.

To validate the effectiveness of $\Phi$, we conduct experiments for CogVideoX 2B (Yang et al., 2025) and `Wan` 1.3B (WanTeam et al., 2025). Compared with the previous SOTA QAT method Q-DM (Li et al., 2023b), the proposed approach exhibits consistently lower $\|\boldsymbol{g}_t\|_2$ and reduced training loss, as depicted in Fig. 3. This aligns well with both the theoretical and the empirical analysis discussed earlier. Therefore, incorporating $\Phi$ in QAT effectively reduces the gradient norm and leads to better convergence for QAT. In addition, as evidenced by Fig. 3, the substantial performance degradation of Q-DM (*e.g.*, depicted

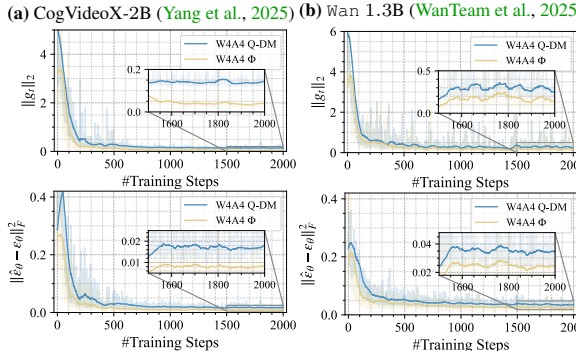

**Figure 3:** (*Upper*) $\|\boldsymbol{g}_t\|_2$ *vs.* #steps and (*Lower*) training loss (*i.e.*, Eq. (6)) *vs.* #steps across different video DMs and 4-bit QAT methods. "$\Phi$" denotes our approach in Sec. 3.1.

in Fig. 1) for the video generation task could be attributed to its relatively large $\|\boldsymbol{g}_t\|_2$. Besides, we provide further analyses of $\|\boldsymbol{g}_t\|_2$ in video generation QAT in Sec. H.

## 3.2 PROGRESSIVELY SHRINKING $\Phi$ VIA *Rank-Decay*

However, during inference, the auxiliary module $\Phi$ introduces non-negligible overhead. Concretely, $\Phi$ incurs additional matrix multiplications between $b$-bit activations $\mathcal{Q}_b(\mathbf{X})$ and full-precision weights $\mathbf{W}_\Phi$. This is inapplicable to low-bit multiplication kernels and thus hinders inference acceleration. In addition, the storage of full-precision $\mathbf{W}_\Phi$ for each $\Phi$ leads to significant memory overhead, exceeding that of the quantized diffusion model by several fold.

To improve QAT while eliminating the inference overhead, we propose to progressively remove $\Phi$ throughout the training process. This allows the model to benefit from $\Phi$ during QAT, while ultimately yielding a standard quantized model (Li et al., 2023b; Esser et al., 2020b) with no extra inference cost. To achieve this goal, a straightforward solution is to decay all parameters of $\Phi$ directly. However, we have noticed that it is ineffective and suboptimal (see Tab. 4). This observation calls for a fine-grained decay strategy.

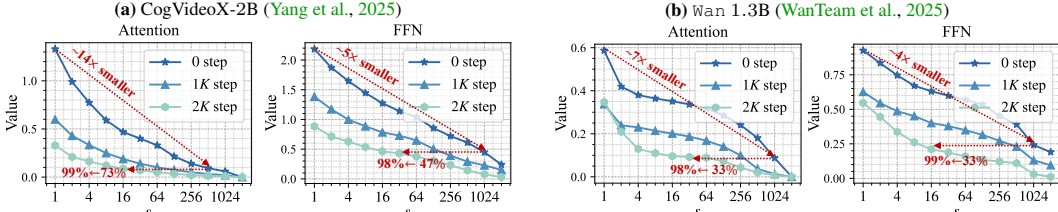

**Figure 4:** Singular value variation in $\mathbf{W}_\Phi$ across training iterations for 4-bit video DMs. We visualize the average of the singular values $\{\sigma_s\}_{s=1,2,\ldots,2^{10}} \cup \{\sigma_d\}$ across layers of all Attention blocks (Vaswani et al., 2023) and feed-forward networks (FFNs), respectively. "0 step" denotes the initialization state before QAT.

Therefore, we begin to investigate the contribution of fine-grained components in $\Phi$. Specifically, we apply singular value decomposition (SVD) (Li et al., 2025; 2023c) to $\mathbf{W}_\Phi$ at various training steps:

$$\mathbf{W}_\Phi = \sum_{s=1}^{d} \sigma_s \boldsymbol{u}_s \boldsymbol{v}_s^\top, \tag{10}$$

where $d = \min\{n, m\}$ and $\sigma_1 \geq \sigma_2 \geq \ldots \geq \sigma_d$ are the singular values. The vectors $\boldsymbol{u}_s$ and $\boldsymbol{v}_s$ denote the left and right singular vectors associated with $\sigma_s$, respectively. By tracking the evolution of the average $\sigma_s$, we observe two key findings (exemplified by Fig. 4 (a) Attention): (*i*) $\mathbf{W}_\Phi$ contains a substantial number of small singular values. For example, approximately $73\%$ (0-th step) of the average $\sigma_s$ are $\sim 14\times$ smaller than the largest one $\sigma_1$; (*ii*) The presence of these small $\sigma_s$ becomes increasingly pronounced as QAT progresses, with the proportion rising to $99\%$ ($2K$-th step).

These findings suggest that an increasing number of orthonormal directions $\{\boldsymbol{u}_s, \boldsymbol{v}_s\}$ contribute little, as their associated singular values $\sigma_s$ are small (Zhang et al., 2015; Yang et al., 2020). We posit that the main weight $\mathbf{W}$ gradually learns to absorb quantization errors. As training proceeds, $\mathbf{W}_\Phi$ collapses to low rank and plays an ever-diminishing role in error compensation. This, in turn, implies that only an increasingly low-rank portion of $\Phi$ (*i.e.*, $\Sigma_{s=1}^{d'} \sigma_s \boldsymbol{u}_s \boldsymbol{v}_s^\top$, where $d' < d$) is needed, and the

remaining components can be decayed without noticeably affecting performance. Motivated by this, we propose a novel *rank-decay* schedule that progressively shrinks $\Phi$ by repeatedly identifying and eliminating the above-mentioned low-impact parts. First, to attain these parts, we reformulate the computation of $\Phi$ as:

$$\Phi(\mathcal{Q}_b(\mathbf{X})) = \mathbf{LR}\mathcal{Q}_b(\mathbf{X}), \tag{11}$$

where $\mathbf{L} = [\sqrt{\sigma_1}\boldsymbol{u}_1, \ldots, \sqrt{\sigma_r}\boldsymbol{u}_r] \in \mathbb{R}^{n \times r}$ and $\mathbf{R} = [\sqrt{\sigma_1}\boldsymbol{v}_1, \ldots, \sqrt{\sigma_r}\boldsymbol{v}_r]^\top \in \mathbb{R}^{r \times m}$ for a given rank $r$. In practice, we set $r \ll d$ to reduce training costs, as $\mathbf{W}_\Phi$ already exhibits a non-negligible number of small singular values before QAT. Consequently, $\sqrt{\sigma_s}\boldsymbol{u}_s$ and $\sqrt{\sigma_s}\boldsymbol{v}_s$ are the components in $\mathbf{W}_\Phi$ represent the $s$-th level of contribution. Then, with a rank-based regularization $\boldsymbol{\gamma}$ applied, the forward computation of a quantized linear layer during training is modified as:

$$\hat{\mathbf{Y}} = \mathcal{Q}_b(\mathbf{W})\mathcal{Q}_b(\mathbf{X}) + (\boldsymbol{\gamma} \odot \mathbf{L})\mathbf{R}\mathcal{Q}_b(\mathbf{X}), \tag{12}$$

where $\boldsymbol{\gamma}$ is defined as:

$$\boldsymbol{\gamma} = \text{concat}([1]_{n \times (1-\lambda)r}, [u]_{n \times \lambda r}) \in \mathbb{R}^{n \times r}. \tag{13}$$

Here, $u$ follows a cosine annealing schedule that decays from 1 to 0, $\lambda \in (0, 1]$ represents the shrinking ratio, and $\odot$ denotes element-wise multiplication. Eq. (12) and Eq. (13) allow us to progressively eliminate the low-impact components of $\Phi$ (*i.e.*, $[\sqrt{\sigma_{(1-\lambda)r+1}}\boldsymbol{u}_{(1-\lambda)r+1}, \ldots, \sqrt{\sigma_r}\boldsymbol{u}_r]$ and $[\sqrt{\sigma_{(1-\lambda)r+1}}\boldsymbol{v}_{(1-\lambda)r+1}, \ldots, \sqrt{\sigma_r}\boldsymbol{v}_r]^\top$). Once $u$ reaches 0, we truncate $\{\mathbf{L}, \mathbf{R}\}$ to $\{\mathbf{L}', \mathbf{R}'\}$, and rewrite $\mathbf{W}_\Phi$ as:

$$\begin{cases} \mathbf{L}' = \mathbf{L}[:, : (1-\lambda)r] \\ \mathbf{R}' = \mathbf{R}[: (1-\lambda)r, :] \end{cases} \Rightarrow \mathbf{W}_\Phi = \mathbf{L}'\mathbf{R}'. \tag{14}$$

In Eq. (14), the rank of $\mathbf{W}_\Phi$ is shrunk from $r$ to $(1-\lambda)r$. During the subsequent training phase, the above procedures (*i.e.*, both decomposition and decay) are iteratively applied. Ultimately, we fully eliminate $\Phi$ by reducing $r$ to 0, which incurs negligible impact on model performance (see Tab. 3). It is worth noting that we set $\lambda = \frac{1}{2}$ for Eq. (13) in this work, based on its effectiveness demonstrated in Tab. 4. Overall, the overview of the *rank-decay* schedule is exhibited in Fig. 2 (b).

## 4 EXPERIMENTS

### 4.1 IMPLEMENTATION DETAILS

**Models.** We conduct experiments on open-source SOTA video DMs, including CogVideoX-2B and 1.5-5B (Yang et al., 2025), and `Wan` 1.3B and 14B (WanTeam et al., 2025). Classifier-free guidance (`CFG`) (Ho & Salimans, 2022) is used for all models, and the frame number of generated videos is fixed to 49 for CogVideoX-2B and 81 for the others.

**Baselines.** We adopt previous powerful PTQ and QAT methods as baselines: ViDiT-Q (Zhao et al., 2025a), SVDQuant (Li et al., 2025), LSQ (Esser et al., 2020b), Q-DM (Li et al., 2023b), and EfficientDM (He et al., 2024). Since these methods were designed for image DMs or convolutional neural networks (CNNs), we adapt these works to video DMs using their open-source code (if available) or the implementation details provided in the corresponding papers. Without specific clarification, static *per-channel* weight quantization with dynamic *per-token* activation quantization, a common practice in the community (Zhao et al., 2025a; Liu et al., 2023), is used for all linear layers.

**Training.** We employ $16K$ captioned videos from `OpenVidHQ-0.4M` (Nan et al., 2025) as the training dataset. The AdamW (Loshchilov & Hutter, 2019) optimizer is utilized with a weight decay of $10^{-4}$. We employ a cosine annealing schedule to adjust the learning rate over training. During QAT, we train `Wan` 14B (WanTeam et al., 2025) and CogVideoX1.5-5B (Yang et al., 2025) for 16 epochs on 32×H100 GPUs and 16×H100 GPUs, respectively. For the other DMs, we employ 8 training epochs on 8×H100 GPUs. Additionally, we allocate the same training iterations for each decay phase (*i.e.*, shrinking the remaining $r$ to $(1-\lambda)r$). The same settings are applied to all QAT baselines.

**Evaluation.** We select 8 dimensions in VBench (Huang et al., 2024c) with unaugmented prompts to comprehensively evaluate the performance following previous studies (Zhao et al., 2025a; Ren et al., 2024). Moreover, for huge ($\geq$ 5B parameters) DMs, we additionally report the results on VBench-2.0 (Zheng et al., 2025a) with augmented prompts to measure the adherence of videos to *physical laws*, *reasoning*, *etc*. More detailed experimental setups can be found in Sec. D.

**Table 1:** Performance comparison across different quantization methods on VBench (Huang et al., 2024c). "†" indicates PTQ methods and "*" signifies QAT methods. "Full Prec." denotes the BF16 model. "♣" represents that we apply fine-grained *per-group* weight-activation quantization with a group size of 64 and keep some linear layers unquantized, which is the same as the official settings of SVDQuant (Li et al., 2025) (details can be found in Sec. D). "Full Fine-tuning" denotes we fine-tune the model with the same data as QVGen. Best and second-best results are highlighted in **bold** and underline formats, respectively.

| Method | #Bits (W/A) | Imaging Quality↑ | Aesthetic Quality↑ | Motion Smoothness↑ | Dynamic Degree↑ | Background Consistency↑ | Subject Consistency↑ | Scene Consistency↑ | Overall Consistency↑ |
|---|---|---|---|---|---|---|---|---|---|
| CogVideoX-2B (CFG = 6.0, 480p, fps = 8) | | | | | | | | | |
| Full Prec. | 16/16 | 59.15 | 54.49 | 97.43 | 67.78 | 94.79 | 92.82 | 36.24 | 25.06 |
| Full Fine-tuning | 16/16 | 61.34 | 56.53 | 98.59 | 65.39 | 93.84 | 93.43 | 34.99 | 25.50 |
| ViDiT-Q (Zhao et al., 2025a)† | 4/6 | 54.72 | 43.01 | 92.18 | 43.22 | 90.76 | 81.02 | 26.25 | 20.41 |
| SVDQuant (Li et al., 2025)† | 4/6 | 58.27 | 47.06 | 95.28 | 40.83 | 92.41 | 87.45 | 27.69 | 21.34 |
| SVDQuant (Li et al., 2025)†♣ | 4/4 | 51.60 | 49.40 | 97.69 | 42.22 | 94.03 | 91.78 | 25.67 | 22.89 |
| LSQ (Esser et al., 2020a)* | 4/4 | 58.73 | 54.20 | 97.57 | 45.00 | 92.97 | 92.41 | 24.06 | 23.17 |
| Q-DM (Li et al., 2023b)* | 4/4 | 54.96 | 52.71 | 98.00 | 48.61 | 93.82 | 91.86 | 28.02 | 23.87 |
| EfficientDM (He et al., 2024)* | 4/4 | 55.96 | 51.97 | 98.03 | 46.67 | 94.10 | 91.70 | 27.76 | 24.28 |
| QVGen (Ours)* | 4/4 | **60.16** | **54.61** | **98.06** | **67.22** | **94.38** | **93.01** | **31.42** | **24.61** |
| LSQ (Esser et al., 2020a)* | 3/3 | 56.46 | 40.35 | 97.98 | 0.56 | 94.08 | 89.18 | 4.80 | 13.80 |
| Q-DM (Li et al., 2023b)* | 3/3 | 50.88 | 40.41 | 98.03 | 5.56 | 93.93 | 87.75 | 7.33 | 15.98 |
| EfficientDM (He et al., 2024)* | 3/3 | 52.86 | 44.58 | 97.13 | 28.61 | 93.15 | 88.26 | 15.42 | 20.42 |
| QVGen (Ours)* | 3/3 | **58.36** | **50.54** | **98.37** | **53.89** | **94.55** | **90.50** | **23.85** | **22.92** |
| Wan 1.3B (CFG = 5.0, 480p, fps = 16) | | | | | | | | | |
| Full Prec. | 16/16 | 64.30 | 58.21 | 97.37 | 70.28 | 95.94 | 93.84 | 28.05 | 24.67 |
| Full Fine-tuning | 16/16 | 64.59 | 58.85 | 97.46 | 83.61 | 94.30 | 93.68 | 27.55 | 24.86 |
| ViDiT-Q (Zhao et al., 2025a)† | 4/6 | 56.24 | 50.18 | 94.81 | 52.43 | 89.67 | 82.53 | 13.45 | 19.58 |
| SVDQuant (Li et al., 2025)† | 4/6 | 58.16 | 51.27 | 97.05 | 49.44 | 93.74 | 91.71 | 14.18 | 23.26 |
| SVDQuant (Li et al., 2025)†♣ | 4/4 | 57.57 | 46.30 | 94.21 | 72.22 | 93.16 | 77.96 | 12.73 | 21.91 |
| LSQ (Esser et al., 2020a)* | 4/4 | 59.11 | 49.09 | **98.35** | 71.11 | 92.66 | 91.67 | 10.38 | 18.83 |
| Q-DM (Li et al., 2023b)* | 4/4 | 60.40 | 52.50 | 97.22 | 76.67 | 93.37 | 89.26 | 13.28 | 21.63 |
| EfficientDM (He et al., 2024)* | 4/4 | 60.70 | 53.57 | 96.18 | 56.39 | 93.74 | 91.70 | 11.77 | 21.19 |
| QVGen (Ours)* | 4/4 | **63.08** | **54.67** | 98.25 | **77.78** | **94.08** | **92.57** | **15.32** | **23.01** |
| LSQ (Esser et al., 2020a)* | 3/3 | 58.80 | 46.86 | 98.22 | 23.61 | 91.86 | 89.42 | 0.89 | 15.51 |
| Q-DM (Li et al., 2023b)* | 3/3 | 56.19 | 44.95 | 95.13 | 76.94 | 92.09 | 83.82 | 1.79 | 16.89 |
| EfficientDM (He et al., 2024)* | 3/3 | 42.32 | 33.52 | 96.50 | 70.28 | 92.10 | 74.79 | 0.04 | 11.38 |
| QVGen (Ours)* | 3/3 | **67.35** | **49.71** | **98.93** | **84.14** | **93.62** | **92.25** | **5.71** | **20.11** |

## 4.2 Performance Analysis

**Comparison with baselines.** We report VBench score comparisons in Tab. 1. In W4A4 quantization, recent QAT methods (Esser et al., 2020b; He et al., 2024; Li et al., 2023b) show non-negligible performance degradation. With W3A3, performance drops become more pronounced. In contrast, the proposed QVGen achieves substantial performance recovery in 3-bit models and comparable results to full-precision models in 4-bit quantization. Specifically, it shows higher scores or less than a 2% decrease in all metrics for W4A4 CogVideoX-2B (Yang et al., 2025), except Scene Consistency. For PTQ baselines (Zhao et al., 2025a; Li et al., 2025), all fail to generate meaningful content in W4A4 *per-channel* and *per-token* settings. Therefore, we apply W4A6 quantization for these methods and also utilize fine-grained *per-group* W4A4 quantization for SVDQuant (Li et al., 2025). In these cases, W4A4 QVGen outperforms them by a large margin, particularly with 8.37 and 14.61 higher Aesthetic Quality and Subject Consistency for Wan 1.3B compared to W4A4 SVDQuant. In addition to these findings, we observe that for Wan 1.3B, Dynamic Degree recovers easily during QAT, even surpassing that of the BF16 model. However, for CogVideoX-2B, this metric significantly drops. Moreover, Scene Consistency is the most challenging metric to maintain across models and methods. Detailed training loss curves across QAT methods and a trial of combining SVDQuant (Li et al., 2025) and QVGen can be found in Sec. G and Sec. I, respectively.

Beyond the quantitative results, we provide qualitative results in Fig. 1 and Sec. P, where QVGen markedly improves visual quality over prior methods. Although a clear gap remains between 3-bit and 4-bit outputs, our approach shifts the Pareto frontier toward higher accuracy at a smaller model size than existing techniques (*e.g.*, 3-bit QVGen for CogVideoX achieves a superior Aesthetic Quality than previous 4-bit methods in Tab. 1). We view this initial exploration as setting a direction and providing valuable insight for future work on 2-bit quantization.

Besides, we provide more comparisons with additional metrics in Sec. J and comparisons under relatively higher bit-width in Sec. E. We also apply our approach to image generation in Sec. M.

**Results for huge DMs.** To demonstrate the scalability of our method, we further test two huge video DMs, including CogVideoX1.5-5B and Wan 14B, at 720p resolution. As illustrated in Tab. 2,

**Table 2:** Performance for huge video DMs on VBench. Comparison with baselines can be found in Sec. K.

| Method | #Bits (W/A) | Imaging Quality↑ | Aesthetic Quality↑ | Motion Smoothness↑ | Dynamic Degree↑ | Background Consistency↑ | Subject Consistency↑ | Scene Consistency↑ | Overall Consistency↑ |
|---|---|---|---|---|---|---|---|---|---|
| CogVideoX1.5-5B (CFG = 6.0, 720p, fps = 16) | | | | | | | | | |
| Full Prec. | 16/16 | 66.25 | 59.49 | 98.42 | 59.72 | 96.57 | 95.28 | 39.14 | 26.18 |
| QVGen (Ours) | 4/4 | 66.76 | 59.52 | 98.38 | 64.44 | 95.83 | 94.88 | 28.47 | 24.45 |
| QVGen (Ours) | 3/3 | 54.44 | 35.85 | 97.23 | 58.89 | 96.48 | 90.17 | 13.27 | 17.15 |
| Wan 14B (CFG = 5.0, 720p, fps = 16) | | | | | | | | | |
| Full Prec. | 16/16 | 67.89 | 61.54 | 97.32 | 70.56 | 96.31 | 94.08 | 33.91 | 26.17 |
| QVGen (Ours) | 4/4 | 66.87 | 59.41 | 97.71 | 76.11 | 96.50 | 94.45 | 19.84 | 25.70 |
| QVGen (Ours) | 3/3 | 48.70 | 29.73 | 99.05 | 93.33 | 97.34 | 94.71 | 2.81 | 13.97 |

our 3-bit and 4-bit models follow the same pattern seen in smaller models (Tab. 1). However, 3-bit quantization incurs much larger drops on demanding metrics such as Scene and Overall Consistency,

underscoring the challenge of pushing these larger models to 3 bits. In Fig. 5, we further assess the models with VBench-2.0; the W4A4 DMs incur only negligible overall performance loss.

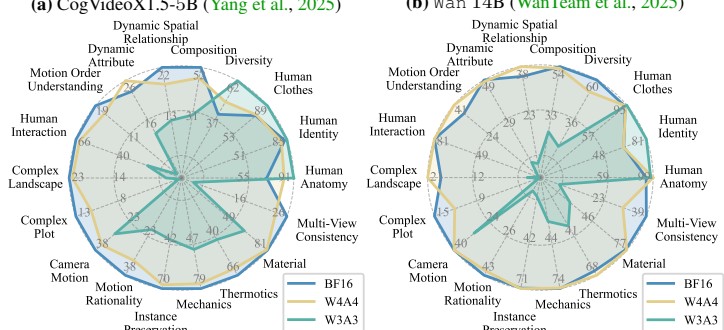

**Figure 5:** Performance for huge video DMs on VBench-2.0 (Zheng et al., 2025a). Our 4-bit models exhibit a minimal drop of ∼1% in total score.

## 4.3 ABLATION STUDIES

To demonstrate the effect of each design, we employ W4A4 Wan 1.3B with VBench (Huang et al., 2024c) for ablation studies. More ablations can be found in Sec. L.

**Effect of different components.** We evaluate the contribution of each component in Tab. 3. The auxiliary module Φ (Sec. 3.1) yields substantial performance improvements across all metrics. Further, the *rank-decay* schedule (Sec. 3.2) effectively eliminates extra inference overhead, while inducing less than a 0.6% drop in most metrics. It even leads to improvement in Overall Consistency.

**Table 3:** Ablation results of each component. "Naive" denotes naive QAT in a KD-based manner. "Rank" signifies our *rank-decay* schedule.

| Method | Imaging Quality↑ | Aesthetic Quality↑ | Dynamic Degree↑ | Scene Consistency↑ | Overall Consistency↑ |
|---|---|---|---|---|---|
| Naive | 60.40 | 52.50 | 76.67 | 13.28 | 21.63 |
| +Φ | **63.41** | **54.75** | **77.89** | **15.51** | 22.98 |
| +Rank | 63.08 | 54.67 | 77.78 | 15.32 | **23.01** |

**Choice of the shrinking ratio λ.** To determine a proper shrinking ratio λ in Eq. (13), we conduct experiments in Tab. 4. By maintaining the same training iterations for each decay phase [5], a small ratio results in an excessively rapid descent of $u$ in Eq. (13) from 1 to 0, potentially destabilizing the training process. On the other hand, a larger ratio may cause the premature removal of high-contributing components during each phase. An extreme scenario would involve a ratio 100% (*i.e.*, λ = 1), in which all $\mathbf{W}_\Phi$ is removed in a single decay phase, leading to huge performance drops.

**Table 4:** Results of different shrinking ratios λ in Eq. (13) for each decay phase. λ = 1 means directly decaying the entire $\mathbf{W}_\Phi$. λ = $\frac{1}{2}$ is used in this work.

| λ | Imaging Quality↑ | Aesthetic Quality↑ | Dynamic Degree↑ | Scene Consistency↑ | Overall Consistency↑ |
|---|---|---|---|---|---|
| 1/4 | 63.02 | 54.23 | 76.84 | 15.18 | 22.85 |
| 1/2 | **63.08** | **54.67** | 77.78 | **15.32** | **23.01** |
| 3/4 | 62.89 | 54.62 | **77.91** | 15.04 | 22.89 |
| 1 | 61.05 | 52.48 | 76.48 | 13.82 | 21.81 |

**Table 5:** Results of different initial ranks r for Eq. (14). r = 0 represents "Naive" in Tab. 3. We employ r = 32 in this work.

| r | Imaging Quality↑ | Aesthetic Quality↑ | Dynamic Degree↑ | Scene Consistency↑ | Overall Consistency↑ |
|---|---|---|---|---|---|
| 0 | 60.40 | 52.50 | 76.67 | 13.28 | 21.63 |
| 8 | 62.71 | 54.47 | 74.62 | 14.42 | 22.81 |
| 16 | 62.99 | 54.62 | 76.58 | 14.84 | 23.00 |
| 32 | **63.08** | **54.67** | **77.78** | 15.32 | **23.01** |
| 64 | 63.06 | 54.30 | 76.74 | **15.40** | 22.92 |

**Choice of the initial rank r.** We further present the results for different initial ranks r of $\mathbf{W}_\Phi$ in Tab. 5. As r increases, the performance gains diminish and eventually deteriorate (*i.e.*, at r = 64). We attribute this trend to the same issue associated with a small shrinking ratio discussed earlier. Specifically, increasing r to 2r introduces an additional decay phase, which shortens the training time allocated to each phase and may lead to overly rapid decay (*e.g.*, at r = 64).

---

[5] We also employ a fixed total number of training epochs.

**Analysis of different fine-grained decay strategies.** To further demonstrate the effectiveness of our *rank-decay* strategy, we evaluate alternative decay strategies in Tab. 6. "Linear" in the table denotes linearly reducing the magnitude of the entire full-rank $\mathbf{W}_\Phi$ to $\mathbf{0}$ by set $\lambda = 1$ with a linear schedule as $u$ for Eq. (13). Inspired by network pruning (Han et al., 2015; 2016), we introduce a "Sparse" strategy that progressively prunes the largest values in $\mathbf{W}_\Phi$ during training. Additionally, motivated by residual quantization (Li et al., 2017), we design a "Res. Q." strategy, which first quantizes $\mathbf{W}_\Phi$ into $4{\times}4$-bit tensors with the same shape and then progressively removes them one by one. Among all these methods, the "Rank" strategy outperforms others across all the metrics by a large margin. Additionally, both "Sparse" and "Res. Q." strategies require at least $1.8\times$ training hours at the same setups compared with our "Rank" approach.

In addition, we introduce two stronger baselines: "Sparse *w/* Wanda" and "Sparse *w/* MaskLLM". Rather than pruning the smallest magnitudes as in "Sparse", the former employs Wanda (Sun et al., 2024) to prune $\mathbf{W}_\Phi$ using 128 randomly selected training samples in each decay phase (see "Sparse" in Sec. F). Following MaskLLM (Fang et al., 2024a), the latter applies learned 2:4 structured pruning masks to $\mathbf{W}_\Phi$ in each phase. All other settings match those of "Sparse". "Sparse *w/* Wanda" yields a small improvement over "Sparse", while "Sparse *w/* MaskLLM" provides a larger gain; however, both remain below "Rank" on all metrics. It is also worth noting that "Sparse *w/* Wanda" requires $1.8\times$ the training time of "Rank", similar to "Sparse", and "Sparse *w/* MaskLLM" requires $2.1\times$ due to mask learning.

**Table 6:** Results of different decay strategies. Details of these methods can be found in Sec. F. "Rank" denotes the *rank-decay* strategy in this work.

| Decay Strategy | Imaging Quality ↑ | Aesthetic Quality ↑ | Dynamic Degree ↑ | Scene Consistency ↑ | Overall Consistency ↑ |
|---|---|---|---|---|---|
| Linear | 60.82 | 52.81 | 73.19 | 13.34 | 21.87 |
| Sparse | 61.15 | 54.06 | 74.24 | 13.86 | 22.52 |
| Sparse *w/* Wanda | 61.43 | 54.08 | 74.36 | 13.94 | 22.48 |
| Sparse *w/* MaskLLM | 61.36 | 54.12 | 74.82 | 14.15 | 22.57 |
| Res. Q. | 61.72 | 54.01 | 72.41 | 14.17 | 22.31 |
| Rank | **63.08** | **54.67** | **77.78** | **15.32** | **23.01** |

## 4.4 Efficiency Discussion

**Inference efficiency.** We report the per-step latency of `W4A4` DiT components on one A800 GPU in Fig. 6 (b). Adapted from the CUDA kernel implementation by Ashkboos et al. (2024), `W4A4` QVGen achieves $1.21\times$ and $1.44\times$ speedups for `Wan 1.3B` and 14B, respectively.

Besides, it exhibits $\sim 4\times$ memory savings compared to the `BF16` format, as shown in Fig. 6 (a). Nevertheless, we believe that the acceleration ratio could be further improved with advanced kernel-fusion techniques and optimization for

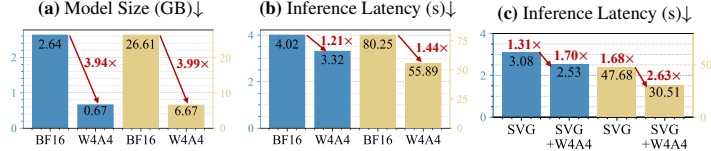

**Figure 6:** Evaluation of memory compression and inference acceleration. Blue color and yellow color denote `Wan 1.3B` and 14B, respectively.

specific tensor shapes in these models, which we leave to future work (More discussion can be found in Secs. N-O). In addition, it is worth noting that our QVGen adheres to standard uniform quantization, enabling drop-in deployment with existing `W4A4` kernels across various devices.

Although previous research (Zhang et al., 2025b) mentioned that the current $3D$ full-attention occupies significant computation in video generation, our QVGen is orthogonal to works that focus on accelerating $3D$ attention and can deliver notable speedups to models that already employ those techniques. For example, when SVG (Xi et al., 2025) is paired with `W4A4` QVGen, the model runs $1.70\times$ and $2.63\times$ faster, compared with $1.31\times$ and $1.68\times$ for SVG alone, as shown in Fig. 6 (c).

**Training efficiency.** Moreover, we conduct a comparative training efficiency analysis across different models and QAT baselines, as demonstrated in Tab. 7. LSQ (Esser et al., 2020b) does not use distillation-based QAT (*i.e.*, no teacher forward pass) and is therefore the fastest to train. Effi-

**Table 7:** Training costs across different methods and models.

| Model | Train. Time (GPU days)↓ | | | | Train. Mem. (GB/GPU)↓ | | | |
|---|---|---|---|---|---|---|---|---|
| | LSQ | Q-DM | EfficientDM | QVGen | LSQ | Q-DM | EfficientDM | QVGen |
| CogVideoX-2B | **8.64** | 9.30 | 8.97 | 9.44 | 62.78 | 67.26 | **44.27** | 67.93 |
| Wan 1.3B | **9.92** | 10.92 | 10.68 | 11.11 | 63.04 | 66.15 | **42.74** | 66.67 |

cientDM (He et al., 2024) updates only LoRA parameters, which substantially reduces optimizer-state memory on GPUs. We believe such a strategy in EfficientDM can be combined with our method to

further improve training efficiency, and we plan to explore this in future work. Relative to distillation-based QAT (*i.e.*, Q-DM (Li et al., 2023b)), our method with low-rank $\Phi$ adds only $\sim 1.02\times$ GPU-days and $\sim 1.01\times$ peak GPU memory for Wan 1.3B model. To be noted, all baselines greatly fall short of our method in final performance, as illustrated in Sec. 4.2. In future work, we will delve into improving QVGen's training efficiency while maintaining its strong performance.

## 5 RELATED WORK

**Video diffusion models.** Building upon the remarkable success of diffusion models (DMs) (Ho et al., 2020a; Song et al., 2021a; Chen et al., 2024a) in image generation (Labs, 2024; Xie et al., 2025), the exploration in the field of video generation (Yang et al., 2025; WanTeam et al., 2025; Kong et al., 2025) is also becoming popular. In contrast to convolution-based diffusion models (Ho et al., 2022; Blattmann et al., 2023), the success of OpenAI Sora (OpenAI, 2024) has spurred researchers to adopt the diffusion transformer (DiT) (Peebles & Xie, 2023) architecture and scale it up for high-quality video generation. However, advanced video DiTs (WanTeam et al., 2025; Yang et al., 2025; Kong et al., 2025) often involve billions of parameters, lengthy multi-step denoising, and intensive computation over long frame sequences. This results in substantial time and memory overhead, which limits their practical deployment. To enable faster video generation, some works have introduced step-distillation (Yin et al., 2025; Lin et al., 2025) on pre-trained models to shorten the denoising trajectory. Others focus on efficient attention (Zhang et al., 2025a; Xi et al., 2025; Huang et al., 2025a), feature caching (Lv et al., 2025; Zou et al., 2025; Huang et al., 2024b), or parallel inference (Fang et al., 2024c;b) to accelerate per-step computations. Moreover, to achieve memory-efficient inference, existing research has explored efficient architecture design (Wu et al., 2024b; Liu et al., 2025), structure pruning (Ben Yahia et al., 2024; Zhao et al., 2025b), and model quantization (Zhao et al., 2025a; Chen et al., 2024b; Tian et al., 2024). These methods aim to achieve both model size and computational cost reduction.

**Model quantization.** Quantization (Jacob et al., 2017) is a predominant technique for minimizing storage and accelerating inference. It can be categorized into post-training quantization (PTQ) (Nagel et al., 2020) and quantization-aware training (QAT) (van Baalen et al., 2020). PTQ compresses models without re-training, making it fast and data-efficient. Nevertheless, it may result in suboptimal performance, especially under ultra-low bit-width (*e.g.*, 3/4-bit). Conversely, QAT applies quantization during training or finetuning and typically achieves higher compression rates with less performance degradation. For DMs, previous quantization research (Li et al., 2023a; He et al., 2023; Huang et al., 2024a; 2025b; So et al., 2023; Wu et al., 2024a; Shang et al., 2023; Wang et al., 2024) primarily focuses on image generation. Video DMs, which incorporate complex temporal and spatial modeling, are still challenging for low-bit quantization. QVD (Tian et al., 2024) and Q-DiT (Chen et al., 2024b) first apply PTQ to convolution-based (Xu et al., 2023b; Guo et al., 2024) and DiT-based (Zheng et al., 2024) video DMs, respectively. Furthermore, ViDiT-Q (Zhao et al., 2025a) employs mixed-precision and fine-grained PTQ to improve performance. However, it experiences video quality loss with 8-bit activation quantization and has not been extended to more advanced models (Yang et al., 2025; WanTeam et al., 2025). Consequently, QAT for advanced video DMs is urgently needed.

In this work, we identify and address the ineffective low-bit QAT for the video DM. Our proposed method significantly enhances model performance and incurs zero inference overhead in 3/4-bit settings. Moreover, our framework is orthogonal to existing QAT methods, which target gradient estimation (Gong et al., 2019), oscillation reduction (Nagel et al., 2022), *etc*.

## 6 CONCLUSIONS AND LIMITATIONS

In this work, we are the first to explore the application of quantization-aware training (QAT) in video DMs. Specifically, we provide a theoretical analysis that identifies that lowering the gradient norm is essential to improve convergence. Then, we propose an auxiliary module ($\Phi$) to achieve this. Additionally, we design a *rank-decay* schedule to progressively eliminate $\Phi$ for zero inference overhead with minimal impact on performance. Extensive experiments for 3-bit and 4-bit quantization validate the effectiveness of our framework, *QVGen*. In terms of limitations, we focus on video generation in this work. However, we believe that our methods can be generalized to more tasks, *e.g.*, natural language processing (NLP), which we will explore in the future.

## ACKNOWLEDGEMENT

This work was supported by the National Natural Science Foundation of China (Nos. 62476018), and the Postdoctoral Fellowship Program of CPSF (No. BX20250487). This work was also supported by the Hong Kong Research Grants Council under the Areas of Excellence scheme grant AoE/E-601/22-R and NSFC/RGC Collaborative Research Scheme grant CRS_HKUST603/22.

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

# Appendix

## CONTENTS

## A  ALOGRITHM OF QVGEN

We summarize our proposed QVGen in Alg. 1. $u$ (see Eq. (13)) in this work follows cosine annealing schedule. $x_0$ and $\mathcal{C}$ denote a clean video clip and its corresponding condition. $N$ is the maximum timestep for training, which is always set to 1000 (Ho et al., 2020b).

---

**Algorithm 1** Procedure of QVGen framework

---

FUNC QVGEN($\mathcal{D}$, $\epsilon_\theta$, it_per_decay_phase, $r$, $\lambda$, $b$)

**Require:** $\mathcal{D}$ — training dataset

        $\epsilon_\theta(\cdot)$ — full-precision DiT model

        it_per_decay_phase — amount of training iterations per decay phase

        $r$ — initial rank of the introduced auxiliary module

        $\lambda$ — shrinking ratio

        $b$ — quantization bit-width

    // Preprocess
1: Standard uniform $b$-bit quantization for $\epsilon_\theta$ to get $\hat{\epsilon}_\theta$          ▷ See Eq. (4)
2: Generate all $\mathbf{W}_\Phi$ for $\hat{\epsilon}_\theta$          ▷ See Eq. (9)
    // Start QAT
3: **while** $r > \frac{1}{\lambda}$ **do:**
4:     Decompose all $\mathbf{W}_\Phi$ to $\{\mathbf{L}, \mathbf{R}\}$ with rank $r$ by Eq. (11)
5:     Generate $\boldsymbol{\gamma}$ by Eq. (13) ▷ $u$ in $\gamma$ will decay from 1 to 0 in following it_per_decay_phase iterations
6:     **for** it in 0 to it_per_decay_phase **do:**
7:         Get batched data pair $(\boldsymbol{x}_0, \mathcal{C})$ from $\mathcal{D}$
8:         $\epsilon \sim \mathcal{N}(\mathbf{0}, \mathbf{I})$
9:         $\tau \sim \text{Uniform}([1, ..., N])$
10:        Generate $\boldsymbol{x}_\tau$ by Eq. (1)
11:        Calculate $\mathcal{L}$ by Eq. (6)          ▷ Eq. (12) is employed in $\hat{\epsilon}_\theta$
12:        Update all $\mathbf{W}$, $s$, $z$, and $\{\mathbf{L}, \mathbf{R}\}$ through back-propagation      ▷ Eq. (5) is applied
13:     **end for**
14:     Truncate $\{\mathbf{L}, \mathbf{R}\}$ to $\{\mathbf{L}', \mathbf{R}'\}$ and regenerate the corresponding $\mathbf{W}_\Phi$ by Eq. (14)
15:     $r = (1 - \lambda)r$
16: **end while**
17: Generate $\boldsymbol{\gamma} = [u]_{n \times r}$      ▷ $u$ in $\gamma$ will decay from 1 to 0 in following it_per_decay_phase iterations
18: Train $\hat{\epsilon}_\theta$ for it_per_decay_phase iterations follow the recipe in Lines 6-13
19: Shrink all $\mathbf{W}_\Phi$ to $\varnothing$      ▷ Since $\gamma = [0]_{n \times r}$ at the end of training, all $\mathbf{W}_\Phi$ can be removed
20: **return** $\hat{\epsilon}_\theta$

---

# B    PROOF OF THM. 3.1

**Assumption B.1.** $f_t$ is convex;

**Assumption B.2.** $\forall \boldsymbol{\theta}_i, \boldsymbol{\theta}_j \in \mathbb{S}^d, \|\boldsymbol{\theta}_i - \boldsymbol{\theta}_j\|_\infty \leq D_\infty$.

**Theorem B.3.** *The average regret is upper-bounded as:* $\frac{R(T)}{T} \leq \frac{dD_\infty^2}{2T\eta_T^m} + \frac{1}{T}\sum_{t=1}^T \frac{\eta_t^M}{2}\|\boldsymbol{g}_t\|_2^2$.

*Proof.* Considering the update for the $p$-th entry of parameters in a quantized video DM:

$$\boldsymbol{\theta}_{t+1,p} = \boldsymbol{\theta}_{t,p} - \eta_{t,p}\boldsymbol{g}_{t,p}, \tag{A}$$

where $\eta_{t,p}$ is the corresponding learning rate, we have:

$$\begin{aligned}(\boldsymbol{\theta}_{t+1,p} - \boldsymbol{\theta}_p^*)^2 &= (\boldsymbol{\theta}_{t,p} - \eta_{t,p}\,\boldsymbol{g}_{t,p} - \boldsymbol{\theta}_p^*)^2 \\ &= (\boldsymbol{\theta}_{t,p} - \boldsymbol{\theta}_p^*)^2 - 2(\boldsymbol{\theta}_{t,p} - \boldsymbol{\theta}_p^*)\eta_{t,p}\boldsymbol{g}_{t,p} + \eta_{t,p}^2\boldsymbol{g}_{t,p}^2.\end{aligned} \tag{B}$$

Rearrange the equation, and divide $2\eta_{t,p}$ on both side as $\eta_{t,p}$ is none-zero,

$$\boldsymbol{g}_{t,p}(\boldsymbol{\theta}_{t,p} - \boldsymbol{\theta}_p^*) = \frac{1}{2\eta_{t,p}}[(\boldsymbol{\theta}_{t,p} - \boldsymbol{\theta}_p^*)^2 - (\boldsymbol{\theta}_{t+1,p} - \boldsymbol{\theta}_p^*)^2] + \frac{\eta_{t,p}}{2}\boldsymbol{g}_{t,p}^2. \tag{C}$$

According to Assm. B.1,

$$f_t(\boldsymbol{\theta}_t) - f_t(\boldsymbol{\theta}^*) \leq \boldsymbol{g}_t^T(\boldsymbol{\theta}_t - \boldsymbol{\theta}^*). \tag{D}$$

Therefore, summing over $d$ dimensions of $\boldsymbol{\theta}$ and $T$ iterations, the regret satisfies:

$$\begin{aligned}R(T) &\leq \sum_{t=1}^T\sum_{p=1}^d \frac{1}{2\eta_{t,p}}[(\boldsymbol{\theta}_{t,p} - \boldsymbol{\theta}_p^*)^2 - (\boldsymbol{\theta}_{t+1,p} - \boldsymbol{\theta}_p^*)^2] + \sum_{t=1}^T\sum_{p=1}^d \frac{\eta_{t,p}}{2}\boldsymbol{g}_{t,p}^2 \\ &= \sum_{p=1}^d [\frac{1}{2\eta_{1,p}}(\boldsymbol{\theta}_{1,p} - \boldsymbol{\theta}_p^*)^2 - \frac{1}{2\eta_{T,p}}(\boldsymbol{\theta}_{T+1,p} - \boldsymbol{\theta}_p^*)^2] \\ &\quad + \sum_{t=2}^T\sum_{p=1}^d \left(\frac{1}{2\eta_{t,p}} - \frac{1}{2\eta_{t-1,p}}\right)(\boldsymbol{\theta}_{t,p} - \boldsymbol{\theta}_p^*)^2 \\ &\quad + \sum_{t=1}^T\sum_{p=1}^d \frac{\eta_{t,p}}{2}\boldsymbol{g}_{t,p}^2.\end{aligned} \tag{E}$$

Considering Assm. B.2, we can further relax the above inequality to:

$$R(T) \leq \sum_{p=1}^{d} \frac{D_\infty^2}{2\eta_{1,p}} + \sum_{t=2}^{T} \sum_{p=1}^{d} \left(\frac{1}{2\eta_{t,p}} - \frac{1}{2\eta_{t-1,p}}\right) D_\infty^2 + \sum_{t=1}^{T} \sum_{p=1}^{d} \frac{\eta_{t,p}}{2} g_{t,p}^2. \tag{F}$$

Denoting the maximum and minimum values for $\{\eta_{t,p}\}_{p=1,\ldots,d}$ as $\eta_t^M$ and $\eta_t^m$, respectively, then:

$$R(T) \leq \frac{dD_\infty^2}{2\eta_T^m} + \sum_{t=1}^{T} \frac{\eta_t^M}{2} \|g_t\|_2^2. \tag{G}$$

Thus, the average regret becomes:

$$\frac{R(T)}{T} \leq \frac{dD_\infty^2}{2T\eta_T^m} + \frac{1}{T} \sum_{t=1}^{T} \frac{\eta_t^M}{2} \|g_t\|_2^2. \tag{H}$$

$\square$

## C CONVERGENCE ANALYSIS WITHOUT REQUIRING CONVEXITY

For completeness, we also provide a nonconvex convergence result for video DMs under standard smoothness assumptions. Here we consider a (possibly nonconvex) training objective $F : \mathbb{S}^d \to \mathbb{R}$ and the gradient descent updates

$$\theta_{t+1} = \theta_t - \eta_t g_t, \qquad g_t := \nabla F(\theta_t), \tag{I}$$

where $t$ indexes the optimization iterations. Regret analysis in Sec. B follows the standard online learning setting, where the per-step loss $f_t$ varies with $t$. In contrast, nonconvex convergence analysis (Bottou et al., 2018; Ghadimi & Lan, 2013) uses a fixed deep-learning objective $F$ across iterations, so no subscript is needed, and convergence is certified by a vanishing minimum gradient norm.

**Assumption C.1** (Smoothness). The function $F$ is $L$-smooth, i.e., for all $\theta, \theta' \in \mathbb{S}^d$,

$$\|\nabla F(\theta) - \nabla F(\theta')\|_2 \leq L\|\theta - \theta'\|_2. \tag{J}$$

Equivalently, $F$ satisfies the standard descent inequality:

$$F(\theta') \leq F(\theta) + \nabla F(\theta)^\top (\theta' - \theta) + \frac{L}{2}\|\theta' - \theta\|_2^2. \tag{K}$$

**Assumption C.2** (Learning rate bounds). The learning rates satisfy

$$0 < \eta^m \leq \eta_t \leq \eta^M \leq \frac{1}{L} \quad \text{for all } t, \tag{L}$$

where $\eta^m$ and $\eta^M$ are the minimum and maximum learning rates[6] across iterations, respectively.

**Assumption C.3** (Lower-bounded objective). There exists $F^\star$ such that $F(\theta) \geq F^\star$ for all $\theta \in \mathbb{S}^d$.

**Theorem C.4** (Convergence to a first-order stationary point). *Under Assms. C.1–C.3, the iterates of Eq. (I) satisfy*

$$\frac{1}{T} \sum_{t=1}^{T} \|g_t\|_2^2 \leq \frac{2\left(F(\theta_1) - F^\star\right)}{T\,\eta^m}. \tag{M}$$

*Consequently,*

$$\min_{1 \leq t \leq T} \|g_t\|_2^2 \leq \frac{1}{T} \sum_{t=1}^{T} \|g_t\|_2^2 \xrightarrow{T \to \infty} 0. \tag{N}$$

*Thus, the iterates converge in the standard nonconvex sense to first-order stationary points, in the sense that there exists an iterate with arbitrarily small gradient norm when $T$ is sufficiently large.*

*Proof.* By $L$-smoothness (the descent inequality) and the update $\theta_{t+1} = \theta_t - \eta_t g_t$,

$$\begin{aligned} F(\theta_{t+1}) &\leq F(\theta_t) + \nabla F(\theta_t)^\top (\theta_{t+1} - \theta_t) + \frac{L}{2}\|\theta_{t+1} - \theta_t\|_2^2 \\ &= F(\theta_t) - \eta_t \|g_t\|_2^2 + \frac{L}{2}\eta_t^2 \|g_t\|_2^2 = F(\theta_t) - \eta_t \left(1 - \frac{L\eta_t}{2}\right)\|g_t\|_2^2. \end{aligned} \tag{O}$$

Since $\eta_t \leq 1/L$, we obtain $1 - L\eta_t/2 \geq 1/2$, hence

$$F(\theta_{t+1}) \leq F(\theta_t) - \frac{\eta_t}{2}\|g_t\|_2^2. \tag{P}$$

---

[6] We employ the same learning rate for the entire quantized DM for simplicity.

Summing over $t \in \{1, \ldots, T\}$ and using $F(\boldsymbol{\theta}_{T+1}) \geq F^\star$ yields

$$\sum_{t=1}^{T} \frac{\eta_t}{2} \|\boldsymbol{g}_t\|_2^2 \leq F(\boldsymbol{\theta}_1) - F^\star. \tag{Q}$$

With $\eta_t \geq \eta^m$, we obtain

$$\frac{1}{T} \sum_{t=1}^{T} \|\boldsymbol{g}_t\|_2^2 \leq \frac{2\left(F(\boldsymbol{\theta}_1) - F^\star\right)}{T\,\eta^m}, \tag{R}$$

which proves Eq. (M). Moreover, we have

$$\min_{1 \leq t \leq T} \|\boldsymbol{g}_t\|_2^2 \ \leq \ \frac{1}{T} \sum_{t=1}^{T} \|\boldsymbol{g}_t\|_2^2 \ \leq \ \frac{2\left(F(\boldsymbol{\theta}_1) - F^\star\right)}{T\,\eta^m} \ \xrightarrow{T \to \infty} \ 0. \tag{S}$$

$\square$

*Remark* C.5 (Connection to our convex analysis). Thm. C.4 shows that, for any finite $T$, the convergence behavior (*i.e.*, Eq. (N)) of QAT is determined by the average gradient norm $\frac{1}{T} \sum_{t=1}^{T} \|\boldsymbol{g}_t\|_2^2$. Since this quantity admits an $\mathcal{O}(1/T)$ upper bound under smoothness and bounded learning rates, reducing $\|\boldsymbol{g}_t\|_2$ during training directly tightens the bound and improves the finite-step convergence of QAT. This matches our convex regret analysis, where the same average gradient norm appears as the core term controlling convergence. Thus, reducing the gradient norm is essential for improving QAT optimization in both settings.

## D  MORE EXPERIMENTAL DETAILS

In this section, we provide additional experimental setups.

**Models.** For testing, we employ DDIM (Song et al., 2021a) and DPM-Solver++ (Lu et al., 2023) for CogVideoX-2B and 1.5-5B models (Yang et al., 2025), respectively. For flow-based `Wan` models (WanTeam et al., 2025), we additionally apply UniPC (Zhao et al., 2023a) corrector and set `flow_shift` (Lipman et al., 2023) to 3.0 and 5.0 for generating $480p$ and $720p$ videos, respectively.

**Baselines.** For QAT baselines, we employ the same settings as our QVGen, *e.g.*, training iterations, batch size, and optimizer, to make a fair comparison. For PTQ baselines, we adopt bit-width in $\{2, \ldots, 8\}$ for the mixed-precision strategy proposed in ViDiT-Q (Zhao et al., 2025a). For `W4A4` group-wise SVDQuant (Li et al., 2025), we retain 16-bit precision for linear layers involved in `adaptive normalization`, `embedding layers`, and the key and value projections in `cross-attention`. For both `W4A4` and `W4A6` SVDQuant, we apply SmoothQuant (Xiao et al., 2024) as a pre-processing step, and GPTQ (Frantar et al., 2023) as a post-processing step. Except as noted above, we only quantize all linear layers for a given video DM. For the attention module, we adopt full-precision `flash-attention` (Dao et al., 2022) to speed up inference, which is a common practice. It is also worth noting that, since we only quantize linear layers and employ `flash-attention` for the attention modules, the "Naive" baseline (see Tab. K) is equivalent to Q-DM (Li et al., 2023b) in this paper.

**Training.** Before QAT, we resize and center-crop the input frames to match the evaluation resolution, except for `Wan` 14B, which uses $480p$ during QAT to reduce training costs. We then sample video clips containing 49 frames at equal-frame intervals. During QAT, `Wan` 14B (WanTeam et al., 2025) and CogVideoX1.5-5B (Yang et al., 2025) are trained using `PyTorch FSDP` (Zhao et al., 2023b), while other diffusion models are trained with `DeepSpeed ZeRO-2` (Rajbhandari et al., 2020). Specifically, we adopt a warm-up phase spanning $\frac{1}{10}$ of the total training epochs, and set the global batch size to 48 for `Wan` 1.3B and 64 for all other models. A cosine annealing schedule is applied to the learning rate [7], initialized at $3 \times 10^{-5}$ for moderate-sized models ($\leq$2B), $5 \times 10^{-5}$ for CogVideoX1.5-5B, and $10^{-5}$ for `Wan` 14B. For weight quantization parameters, we adopt LSQ (Esser et al., 2020b) with an initial learning rate of $3 \times 10^{-5}$. As described in the preliminary section of the main text, we use the straight-through estimator (STE) (Bengio et al., 2013) to ensure the differentiability of the quantization process. To be noted, when the remaining rank $r < \frac{1}{\lambda}$, we directly apply a cosine annealing function (*i.e.*, $\boldsymbol{\gamma} = [u]_{n \times r}$) to gradually shrink the remaining $\mathbf{W}_\Phi$ to $\varnothing$. Moreover, the training days are summarized in Tab. A.

**Evaluation.** For VBench (Huang et al., 2024c), we test $1\sim 2$B models on $8 \times$H100 GPUs. For huge DMs, we evaluate CogVideoX1.5-5B on $64 \times$H100 GPUs and `Wan` 14B on $128 \times$H100 GPUs for

---

[7]This mentioned learning rate is applied to both model weights and the introduced $\Phi$.

**Table A:** #GPU days (H100) across different models. We present the increased time of QVGen compared with the naive QAT in a KD-based manner in red subscripts. QVGen, which uses $r = 32$ in this paper, incurs negligible time overhead but significantly higher performance (see Tab. K) than the naive method.

| Method/Model | CogVideoX-2B | Wan 1.3B | CogVideoX1.5-5B | Wan 14B |
|---|---|---|---|---|
| QVGen (Ours) | $9.44_{+0.14}$ | $11.11_{+0.18}$ | $\sim51$ | $\sim182$ |

both VBench and VBench-2.0 (Zheng et al., 2025a). The batch size is set to one per GPU, and each run completes within one day. Besides, we sample 5 videos for each unaugmented text prompt across the 8 dimensions in VBench. For VBench-2.0 (Zheng et al., 2025a), we generate 3 videos per augmented text prompt across all dimensions, except for the Diversity dimension, where we generate 20 videos for each prompt.

# E    COMPARISON WITH BASELINES UNDER RELATIVELY HIGHER BIT-WIDTH

Besides 3/4-bit settings, we also conduct experiments under W6A6. As shown in Tab. B, our QVGen achieves full-precision comparable or even better performance than BF16 models. Since these settings are not challenging enough for QAT baselines, which also demonstrate satisfactory performance, our QVGen only shows moderate improvements.

**Table B:** Performance comparison across different methods on VBench under W6A6 quantization for Wan 1.3B.

| Method | Imaging Quality↑ | Aesthetic Quality↑ | Dynamic Degree↑ | Scene Consistency↑ | Overall Consistency↑ |
|---|---|---|---|---|---|
| Full Prec. | 64.30 | 58.21 | 70.28 | 28.05 | 24.67 |
| SVDQuant (Li et al., 2025) | 62.05 | 54.37 | 71.08 | 23.25 | 24.56 |
| LSQ (Esser et al., 2020b) | 63.20 | **57.83** | 76.33 | 24.86 | 24.07 |
| Q-DM (Li et al., 2023b) | 62.89 | 56.24 | 74.64 | 25.38 | 25.12 |
| EfficientDM (He et al., 2024) | 63.38 | 56.42 | 68.68 | 21.47 | 23.57 |
| QVGen (Ours) | **64.27** | 57.69 | **78.02** | **26.84** | **25.53** |

# F    ADDITIONAL FINE-GRAINED DECAY STRATEGIES

In this section, we detail the "Sparse" and "Res. Q." decay strategies (see Tab. N), which shrink $\Phi$ to $\varnothing$, as follows:

- *"Sparse" strategy.* Inspired by pruning techniques (Han et al., 2015; 2016), at the start of QAT, we sparsify $\mathbf{W}_\Phi$ with a sparse ratio of $a\% = 50\%$. Specifically, we set the $a\%$ smallest-magnitude values in $\mathbf{W}_\Phi$ to zero and freeze them in the whole training process. The quantized model is then trained with the remaining non-zero values in $\mathbf{W}_\Phi$. We divide the training process into 6 equal-length phases. In each subsequent phase, we set an additional $\frac{a}{2}\%$ of the remaining non-zero values in $\mathbf{W}_\Phi$ to zero and update $a \leftarrow \frac{a}{2}$. The resulting cumulative sparse ratios across the 6 phases are as follows: $50\%_{(1\text{-st phase})} \rightarrow 75\%_{(2\text{-nd phase})} \rightarrow \cdots \rightarrow 96.875\%_{(5\text{-th phase})} \rightarrow 100\%_{(6\text{-th phase})}$.

- *"Res. Q" strategy.* Motivated by residual quantization (Li et al., 2017), we first decompose $\mathbf{W}_\Phi$ into a series of 4-bit quantized residuals. At the beginning of QAT, we quantize $\mathbf{W}_\Phi$ to its 4-bit approximation $\mathcal{Q}_4(\mathbf{W}_\Phi)$, and then recursively quantize the quantization-induced residuals. Specifically, we define $\mathbf{E}_1 = \mathbf{W}_\Phi - \mathcal{Q}_4(\mathbf{W}_\Phi)$ and quantize it to $\mathcal{Q}_4(\mathbf{E}_1)$; the remaining residual $\mathbf{E}_2 = \mathbf{E}_1 - \mathcal{Q}_4(\mathbf{E}_1)$ is quantized to $\mathcal{Q}_4(\mathbf{E}_2)$; and finally, $\mathbf{E}_3 = \mathbf{E}_2 - \mathcal{Q}_4(\mathbf{E}_2)$ is quantized to $\mathcal{Q}_4(\mathbf{E}_3)$. This yields a 4-term additive decomposition of $\mathbf{W}_\Phi$:

$$\mathbf{W}_\Phi = \mathcal{Q}_4(\mathbf{W}_\Phi) + \mathcal{Q}_4(\mathbf{E}_1) + \mathcal{Q}_4(\mathbf{E}_2) + \mathcal{Q}_4(\mathbf{E}_3). \tag{T}$$

These 4×4-bit quantized tensors[8] are jointly trained with the rest of the quantized model. During QAT, we divide the training process into 4 equal-length phases, and apply cosine annealing within each phase to progressively decay these components in the following order: $\mathcal{Q}_4(\mathbf{E}_3)_{(1\text{-st phase})} \rightarrow \mathcal{Q}_4(\mathbf{E}_2)_{(2\text{-nd phase})} \rightarrow \mathcal{Q}_4(\mathbf{E}_1)_{(3\text{-rd phase})} \rightarrow \mathcal{Q}_4(\mathbf{W}_\Phi)_{(4\text{-th phase})}$.

Compared with our *rank-decay* strategy (with $r = 32$), both alternative methods require storing multiple tensors[9], each with the same shape as the corresponding linear layer's weight matrix. In particular, the "Res. Q." strategy incurs substantial memory overhead and necessitates CPU

---

[8]The use of these 4×4-bit tensors is designed to align with the original BF16 format, as 4×4-bit data could theoretically match a 16-bit representation.

[9]The "Sparse" strategy additionally requires storing binary masks to enforce sparsity.

offloading to avoid out-of-memory (OOM) issues. Furthermore, the "Sparse" strategy introduces extra computational cost due to the matrix multiplication between the mask and $\mathbf{W}_\Phi$. In practice, the "Sparse" and "Res. Q." strategies consume 20.12 and 28.78 GPU days, respectively, whereas our proposed *rank-decay* requires only 11.11 GPU days (see Tab. A). More importantly, *rank-decay* also achieves significantly better performance than both alternatives, as demonstrated in Tab. N.

## G  TRAINING LOSS CURVES

In this section, we present training loss curves across different methods and models. As shown in Fig. A, QVGen and our method in Sec. 3.1 achieve faster and more stable convergence, which supports the effectiveness of the proposed approaches. Note that LSQ (Esser et al., 2020b) uses $\mathbb{E}_{\boldsymbol{x}_0,\mathcal{C},\tau}[\|\boldsymbol{\epsilon} - \boldsymbol{\epsilon}_\theta(\boldsymbol{x}_\tau,\mathcal{C},\tau)\|_F^2]$ with $\boldsymbol{x}_\tau = \alpha_\tau \boldsymbol{x}_0 + \sigma_\tau \boldsymbol{\epsilon}$ as its training objective, instead of Eq. (6) used by the remaining distillation-based methods.

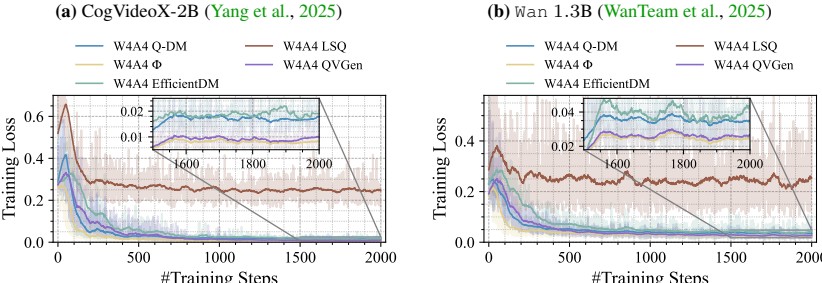

**Figure A:** Training loss *vs.* #steps across different video DMs and 4-bit QAT methods. "Φ" denotes our approach in Sec. 3.1

## H  FURTHER ANALYSES OF GRADIENT NORM IN VIDEO GENERATION QAT

### H.1  VIDEO GENERATION QAT *vs.* IMAGE GENERATION QAT

First, we conduct experiments to compare the gradient norm between image generation QAT and video generation QAT. In Tab. C, we employ W4A4 Q-DM (Li et al., 2023b) to quantize the diffusion models under the same settings. We observe that reducing the gradient norm during QAT, ignored by previous research (Li et al., 2023b; He et al., 2024; Esser et al., 2020b), is far more critical for video generation QAT than for image generation QAT. As Tab. C shows, with a similar parameter count and the same QAT recipe, video diffusion reaches a significantly larger gradient norm than image diffusion. This leads to much more unstable optimization (Xie et al., 2024) in training. We believe the phenomenon happens because video generation introduces complicated temporal modeling, which eventually makes quantization for video diffusion more challenging than image diffusion.

**Table C:** Average gradient norm comparison. SD3-medium (Esser et al., 2024a) is an advanced diffusion model for image generation. We train SD3-medium for $2K$ steps with $16K$ images from the LAION-5B dataset (Schuhmann et al., 2022) on $8\times$H100 GPUs. "Avg. $\|\boldsymbol{g}_t\|_2$" is the average of the gradient norm across training steps.

| Model | SD3-medium | CogVideoX-2B |
|---|---|---|
| Avg. $\|\boldsymbol{g}_t\|_2$ | **0.2047** | 0.3283 |
| #Params. (B) | 2.0 | 2.0 |

### H.2  IMPACT OF MOTION DYNAMICS ON GRADIENT NORM

Here, we study how the motion dynamics of video generation affect the gradient norm. Specifically, using UniMatch (Xu et al., 2023a), we compute an optical-flow score as a motion difference score for each clip and split the training data (Sec. 4.1) into high-motion and low-motion subsets, each with $8K$ videos. In Tab. D, the model trained on the low-motion subset shows a much lower average gradient norm and a better Imaging Quality score, but its Dynamic Degree is lower than that of the model

trained on the high-motion subset. This pattern suggests that a high-motion training set makes QAT harder and less stable (*i.e.*, higher gradient norm), lowering static quality but boosting motion quality. The reverse holds for low-motion clips. To be

**Table D:** Results between different training videos. We employ `W4A4` QVGen with the same configurations as those in Tab. 1 to quantize CogVideoX-2B (Yang et al., 2025).

| $8K$ Training Videos | Avg. $\|\boldsymbol{g}_t\|_2$ | Imaging Quality↑ | Dynamic Degree↑ |
|---|---|---|---|
| high-motion | 0.1972 | 57.47 | **68.21** |
| low-motion | **0.1768** | **60.63** | 62.54 |
| half high-motion + half low-motion | 0.1821 | 60.12 | 67.08 |

noted, with a mixed dataset (containing low-motion + high-motion) in the 4-th row of Tab. D, the quantized model generalizes well in producing either high-motion or low-motion content. Therefore, although a low-motion dataset in training can cause slightly lower $\|\boldsymbol{g}_t\|_2$ (*i.e.*, better training convergence), it is necessary to properly add high-motion data to improve motion quality.

### H.3 DIRECTLY REGULATE THE GRADIENT NORM

In this subsection, we study the effect of directly constraining the gradient norm during QAT. We use `torch.nn.utils.clip_grad_norm_` to rescale the gradients. As shown in Tab. E, reducing the clipping threshold from $1.0$ to $0.5$ improves performance, which supports the benefit of controlling gradient norms for video generation QAT. However, a threshold of $0.1$ causes clear performance degradation, likely because the quantized model is updated too weakly or aggressive gradient clipping disrupts normal QAT. This highlights the need for more principled ways to reduce the gradient norm.

**Table E:** `W4A4` results for `Wan 1.3B` under different thresholds for gradient clipping. "1.0" corresponds to our baseline Q-DM (Li et al., 2023b). We use a clipping threshold of $1.0$ for all other experiments in the paper.

| Grad. Clipping | Imaging Quality↑ | Aesthetic Quality↑ | Dynamic Degree↑ | Scene Consistency↑ | Overall Consistency↑ |
|---|---|---|---|---|---|
| 1.0 | 60.40 | 52.50 | **76.67** | **13.28** | 21.63 |
| 0.5 | **60.58** | **52.95** | 76.50 | **13.28** | **21.71** |
| 0.1 | 56.68 | 51.06 | 71.00 | 12.78 | 21.42 |

## I COMBINATION WITH SVDQUANT

In Tab. F, we show that our method can be combined with the current SOTA PTQ method SVDQuant (Li et al., 2025). We first apply SVDQuant to obtain a weight-modified DM, the quantization parameters, and the low-rank matrices, which we reuse as $\Phi$. We then run QVGen, which progressively removes $\Phi$. This combination yields further gains, likely due to the strong initialization from SVDQuant. We also evaluate an option that updates only the quantization parameters and $\Phi$ within this combination. It achieves sizable improvements over SVDQuant alone, but it still falls short of QVGen when model weights are updated. This suggests that QAT, which trains model weights, is currently important for video generation quantization. We will continue to explore more efficient ways to quantize video DMs while preserving performance.

**Table F:** `W4A4` results of the combination with SVDQuant (Li et al., 2025) for `Wan 1.3B` (WanTeam et al., 2025). "♡" denotes we freeze the weights of the DM and only finetune the quantization parameters and the introduced $\Phi$. "♣" means we employ a more fine-grained and performance-friendly quantization setting as in SVDQuant's paper (details can be found in Sec. D).

| Method | Imaging Quality↑ | Aesthetic Quality↑ | Dynamic Degree↑ | Scene Consistency↑ | Overall Consistency↑ |
|---|---|---|---|---|---|
| Full Prec. | 64.30 | 58.21 | 70.28 | 28.05 | 24.67 |
| SVDQuant♣ (Li et al., 2025) | 57.57 | 46.30 | 72.22 | 12.73 | 21.91 |
| QVGen | 63.08 | 54.67 | **77.78** | 15.32 | 23.01 |
| QVGen *w/* SVDQuant (Li et al., 2025) | **63.64** | **56.23** | 77.42 | **17.65** | **23.89** |
| QVGen♡ *w/* SVDQuant (Li et al., 2025) | 61.38 | 52.76 | 75.85 | 14.12 | 22.47 |

## J COMPARISON WITH BASELINES ON ADDITIONAL METRICS

We also evaluate the similarity between videos generated by different quantization methods and those generated by `BF16` models on VBench (Huang et al., 2024c) captions. Specifically, we employ PSNR (Peak Signal-to-Noise Ratio), SSIM (Structural Similarity) (Wang et al., 2004), and LPIPS (Learned Perceptual Image Patch Similarity) (Zhang et al., 2018). In Tab. G, QVGen substantially outperforms the baselines on these metrics.

**Table G:** Additional `W4A4` performance comparison across different quantization methods. We employ the same models as those in Tab. 1.

| Method | CogVideoX-2B | | | Wan 1.3B | | |
|---|---|---|---|---|---|---|
| | PSNR↑ | SSIM↑ | LPIPS↓ | PSNR↑ | SSIM↑ | LPIPS↓ |
| SVDQuant (Li et al., 2025) | 11.06 | 0.3829 | 0.6305 | 10.14 | 0.3595 | 0.6907 |
| LSQ (Esser et al., 2020b) | 11.75 | 0.4158 | 0.6187 | 11.65 | 0.4743 | 0.6235 |
| Q-DM (Li et al., 2023b) | 12.07 | 0.4270 | 0.6240 | 11.22 | 0.4657 | 0.5942 |
| EfficientDM (He et al., 2024) | 11.91 | 0.4387 | 0.6220 | 11.29 | 0.3926 | 0.6232 |
| QVGen (Ours) | **16.74** | **0.6085** | **0.4127** | **15.94** | **0.5782** | **0.4887** |

## K  COMPARISON WITH BASELINES FOR HUGE DMS

We include a comparison on VBench for large-scale CogVideoX-1.5 5B in Tab. H. QVGen again surpasses all baselines, just as it does on smaller models, underscoring its strength across a wide range of model sizes. Limited resources currently prevent us from adding more baselines for these large-scale models, but we plan to do so in future work.

**Table H:** Performance comparison for huge DMs across different methods on VBench (Huang et al., 2024c). We employ `W4A4` CogVideo-X 5B (Yang et al., 2025) here.

| Method | Imaging Quality↑ | Aesthetic Quality↑ | Dynamic Degree↑ | Scene Consistency↑ | Overall Consistency↑ |
|---|---|---|---|---|---|
| Full Prec. | 61.15 | 54.06 | 74.24 | 13.86 | 22.52 |
| Q-DM (Li et al., 2023b) | 61.72 | 54.01 | 72.41 | 14.17 | 22.31 |
| EfficientDM (He et al., 2024) | 61.72 | 54.01 | 72.41 | 14.17 | 22.31 |
| QVGen (Ours) | **63.08** | **54.67** | **77.78** | **15.32** | **23.01** |

## L  MORE ABLATION STUDIES

### L.1  ROBUSTNESS OF *Rank-Decay* SCHEDULE ACROSS DIFFERENT ANNEALING FUNCTIONS

In this section, we test $5$ different annealing functions for $u$. The results in Tab. I reveal that all the functions yield comparable performance, highlighting the robustness of our approach.

**Table I:** Results of different annealing factors $u$. All of them decay from $1$ to $0$. We employ "Cosine" in this work.

| $u$ | Imaging Quality↑ | Aesthetic Quality↑ | Motion Smoothness↑ | Dynamic Degree↑ | Background Consistency↑ | Subject Consistency↑ | Scene Consistency↑ | Overall Consistency↑ |
|---|---|---|---|---|---|---|---|---|
| Cosine | 63.08 | **54.67** | 98.25 | 77.78 | 94.08 | 92.57 | **15.32** | **23.01** |
| Logarithmic | **63.15** | 54.46 | 98.02 | 77.52 | 93.98 | **92.59** | 14.99 | 22.87 |
| Exponential | 63.04 | 54.48 | 97.88 | **77.81** | 93.96 | 92.56 | **15.32** | 22.66 |
| Square | 63.02 | 54.59 | **98.41** | 77.24 | **94.12** | 92.57 | 15.18 | 22.94 |
| Linear | 63.10 | 54.63 | 98.31 | 77.44 | 94.06 | 92.58 | 15.24 | 22.91 |

### L.2  INITIALIZATION METHODS FOR AUXILIARY MODULES $\Phi$

Besides employing $\mathbf{W} - \mathcal{Q}_b(\mathbf{W})$ to initialize $\mathbf{W}_\Phi$, we provide an alternative initialization approach that considers both weight and activation effects. Specifically, we train each $\mathbf{W}_\Phi$ for 200 iterations to minimize the quantization error of its corresponding linear layer's output (*i.e.*, $\arg \min_{\mathbf{W}_\Phi} \|\mathbf{Y} - \hat{\mathbf{Y}}\|_F^2$). The resulting layer-wise trained $\mathbf{W}_\Phi$ is then used to initialize $\Phi$. As shown in Tab. J, both initialization strategies yield similar performance. Therefore, we believe that QVGen is not sensitive to such choices of initialization. Moreover, we also consider two deliberately suboptimal initialization approaches: zero initialization (*i.e.*, "$\mathbf{0}$") and initialization with parameters randomly sampled from a normal distribution (*i.e.*, "Random"). The "$\mathbf{0}$" scheme leads to a slight performance degradation, while "Random" causes a more noticeable performance drop. We attribute this to the fact that "$\mathbf{0}$" does not compensate for quantization errors, whereas "Random" further injects noise into the model.

### L.3  COMPLETE RESULTS OF TABLES IN ABLATION STUDIES

In Tabs. K to N, we present the complete ablation results across all $8$ dimensions on VBench (Huang et al., 2024c), corresponding to the incomplete versions shown in the ablation study of the main text. These results are consistent with the analyses provided in the main text.

**Table J:** `W4A4` results of different initialization strategies for $\mathbf{W}_\Phi$.

| Init. Strategy | Imaging Quality ↑ | Aesthetic Quality ↑ | Dynamic Degree ↑ | Scene Consistency ↑ | Overall Consistency ↑ |
|---|---|---|---|---|---|
| CogVideoX-2B (CFG = 6.0, 480p, fps = 8) | | | | | |
| $\mathbf{W} - \mathcal{Q}_b(\mathbf{W})$ | **60.16** | 54.61 | **67.22** | **31.42** | 24.61 |
| Layer-wise Train. | 59.97 | **54.84** | 66.71 | 31.14 | **25.02** |
| **0** | 59.86 | 54.47 | 65.59 | 31.32 | 24.59 |
| Random | 49.42 | 37.68 | 26.57 | 6.24 | 11.68 |
| Wan 1.3B (CFG = 5.0, 480p, fps = 16) | | | | | |
| $\mathbf{W} - \mathcal{Q}_b(\mathbf{W})$ | 63.08 | **54.67** | **77.78** | 15.32 | **23.01** |
| Layer-wise Train. | 63.23 | **54.67** | 77.56 | **15.38** | 23.00 |
| **0** | 62.80 | 54.59 | 77.69 | 15.28 | 22.98 |
| Random | 54.41 | 44.14 | 34.44 | 3.13 | 10.17 |

**Table K:** Complete ablation results of each component. "Naive" denotes naive QAT in a KD-based manner. "$-decay$" denotes the setting where $\mathbf{W}_\Phi = \mathbf{LR}$ is initialized with $r = 32$ but not eliminated during QAT. The comparable performance between "$-$Decay" and "$+\Phi$" validates that a low-rank setting with $r < d$ (as mentioned in the main text) is sufficient. Furthermore, the negligible performance loss of "$+$Rank" compared to "$-$Decay" confirms the effectiveness of our proposed decay strategy.

| Method | Imaging Quality ↑ | Aesthetic Quality ↑ | Motion Smoothness ↑ | Dynamic Degree ↑ | Background Consistency ↑ | Subject Consistency ↑ | Scene Consistency ↑ | Overall Consistency ↑ |
|---|---|---|---|---|---|---|---|---|
| Naive | 60.40 | 52.50 | 97.22 | 76.67 | 93.37 | 89.26 | 13.28 | 21.63 |
| $+\Phi$ | **63.41** | **54.75** | **98.40** | **77.89** | **94.36** | 93.29 | **15.51** | 22.98 |
| $+$Rank | 63.08 | 54.67 | 98.25 | 77.78 | 94.08 | 92.57 | 15.32 | **23.01** |
| $-$Decay | 63.32 | 54.64 | 98.34 | 77.79 | 94.15 | 92.61 | 15.40 | 23.03 |

**Table L:** Complete results of different shrinking ratios $\lambda$ for each decay phase. $\lambda = 1$ means directly decaying the entire $\mathbf{W}_\Phi$.

| $\lambda$ | Imaging Quality ↑ | Aesthetic Quality ↑ | Motion Smoothness ↑ | Dynamic Degree ↑ | Background Consistency ↑ | Subject Consistency ↑ | Scene Consistency ↑ | Overall Consistency ↑ |
|---|---|---|---|---|---|---|---|---|
| 1/4 | 63.02 | 54.23 | 97.89 | 76.84 | 94.02 | 92.13 | 15.18 | 22.85 |
| 1/2 | **63.08** | **54.67** | **98.25** | 77.78 | **94.08** | **92.57** | **15.32** | **23.01** |
| 3/4 | 62.89 | 54.62 | 98.15 | **77.91** | 93.89 | 91.63 | 15.04 | 22.89 |
| 1 | 61.05 | 52.48 | 97.31 | 76.48 | 93.42 | 90.04 | 13.82 | 21.81 |

**Table M:** Complete results of different initial ranks $r$. $r = 0$ represents "Naive" in Tab. K.

| $r$ | Imaging Quality ↑ | Aesthetic Quality ↑ | Motion Smoothness ↑ | Dynamic Degree ↑ | Background Consistency ↑ | Subject Consistency ↑ | Scene Consistency ↑ | Overall Consistency ↑ |
|---|---|---|---|---|---|---|---|---|
| 0 | 60.40 | 52.50 | 97.22 | 76.67 | 93.37 | 89.26 | 13.28 | 21.63 |
| 8 | 62.71 | 54.47 | 97.95 | 74.62 | 93.76 | 91.05 | 14.42 | 22.81 |
| 16 | 62.99 | 54.62 | **98.31** | 76.58 | 93.92 | 91.82 | 14.84 | 23.00 |
| 32 | **63.08** | **54.67** | 98.25 | **77.78** | **94.08** | **92.57** | 15.32 | **23.01** |
| 64 | 63.06 | 54.30 | 98.18 | 76.74 | 94.01 | 91.49 | **15.40** | 22.92 |

**Table N:** Complete results of different decay strategies. "Rank" denotes the *rank-decay* strategy in this work. To be noted, the "Sparse" and "Res. Q." strategies incur substantially $1.81\times$ and $2.60\times$ GPU days for training compared with the "Rank" approach, respectively (see Sec. F).

| Decay Strategy | Imaging Quality ↑ | Aesthetic Quality ↑ | Motion Smoothness ↑ | Dynamic Degree ↑ | Background Consistency ↑ | Subject Consistency ↑ | Scene Consistency ↑ | Overall Consistency ↑ |
|---|---|---|---|---|---|---|---|---|
| Sparse | 61.15 | 54.06 | 97.45 | 74.24 | 93.32 | 90.63 | 13.86 | 22.52 |
| Res. Q. | 61.72 | 54.01 | 97.62 | 72.41 | 93.46 | 91.24 | 14.17 | 22.31 |
| Rank | **63.08** | **54.67** | **98.25** | **77.78** | **94.08** | **92.57** | **15.32** | **23.01** |

### L.4 Additional Rank-Based Regularization $\gamma$

In this section, we conduct experiments to validate the superiority of the proposed rank-based regularization compared with additional rank-based regularization. As shown in Tab. O, the setting "concat($[1]_{n \times (1-\lambda)r}, [u]_{n \times \lambda r}$)" (adopted in this work) outperforms both "Random$\times 3$" and "concat($[u]_{n \times \lambda r}, [1]_{n \times (1-\lambda)r}$)" by a large margin. Moreover, the results in the table confirm our idea that removing $\Phi$ by repeatedly decaying components of $\mathbf{W}_\Phi$ associated with small singular values maintains performance. This also reflects that components associated with small singular values contribute little (Zhang et al., 2015; Yang et al., 2020) under the setting of this paper (*i.e.*, jointly training $\Phi$ and the quantized video DM during QAT).

**Table O:** Results of different $\gamma$. "concat($[u]_{n \times \lambda r}, [1]_{n \times (1-\lambda)r}$)" denotes the setting where components of $\mathbf{W}_\Phi$ associated with the largest singular values are decayed in each phase. "Random×3" represents the average performance over 3 experiments, each employing a different randomly generated $\gamma$ per decay phase. Specifically, a random index set $\mathcal{S}_u = \{b_1, b_2, \ldots, b_{\lambda r}\}$ is sampled such that $b_i \in \{1, 2, \ldots, r\}$ and $\forall i \neq j, b_i \neq b_j$. We then define the complementary index set as $\mathcal{S}_1 = \{1, 2, \ldots, r\} \setminus \mathcal{S}_u$. The decay matrix $\gamma \in \mathbb{R}^{n \times r}$ is constructed by setting $\gamma_{:, \mathcal{S}_u} = [u]_{n \times \lambda r}$ and $\gamma_{:, \mathcal{S}_1} = [1]_{n \times (1-\lambda)r}$. The green subscripts indicate the standard deviations.

| $\gamma$ | Imaging Quality↑ | Aesthetic Quality↑ | Motion Smoothness↑ | Dynamic Degree↑ | Background Consistency↑ | Subject Consistency↑ | Scene Consistency↑ | Overall Consistency↑ |
|---|---|---|---|---|---|---|---|---|
| concat($[1]_{n \times (1-\lambda)r}, [u]_{n \times \lambda r}$) | **63.08** | **54.67** | **98.25** | **77.78** | **94.08** | **92.57** | **15.32** | **23.01** |
| Random ×3 | 60.96$_{\pm 0.41}$ | 53.13$_{\pm 0.67}$ | 97.40$_{\pm 0.24}$ | 76.45$_{\pm 0.08}$ | 93.40$_{\pm 0.03}$ | 90.76$_{\pm 0.30}$ | 13.77$_{\pm 0.22}$ | 21.92$_{\pm 0.51}$ |
| concat($[u]_{n \times \lambda r}, [1]_{n \times (1-\lambda)r}$) | 60.56 | 52.61 | 97.28 | 75.36 | 93.35 | 89.47 | 13.46 | 22.24 |

## L.5 Duration of Each Decay for $\gamma$

Here, we discuss the situation if $\Phi$'s rank is diminished too rapidly relative to the schedule $\gamma$. In this case, we believe that keeping the redundant parts (that is, the rank-diminished components in $\Phi$) during QAT does not harm performance and can even lead to a small improvement. This is mainly because the slow change of $u$ in $\gamma$ ($1 \to 0$) increases the training time by making each decay phase longer. The results in the following table support this intuition.

**Table P:** `W4A4` quantization results across different durations of each decay phase for Wan 1.3B. We control the duration to determine the changing speed of the schedule $\gamma$. When applying a long duration, we suggest $\Phi$'s rank is diminished (as an intrinsic behavior) rapidly relative to the schedule $\gamma$.

| Duration (Epoch) | Imaging Quality↑ | Aesthetic Quality↑ | Dynamic Degree↑ | Scene Consistency↑ | Overall Consistency↑ |
|---|---|---|---|---|---|
| 3/4 | 63.08 | 54.67 | 77.78 | 15.32 | 23.01 |
| 1 | 63.07 | **55.09** | 77.54 | 15.29 | **23.03** |
| 3/2 | **63.11** | 54.58 | **78.15** | **15.36** | 23.01 |

## L.6 Weight-Only Quantization *vs.* Activation-Only Quantization

As demonstrated in Fig. B, activation-only quantization causes severe degradation in video generation quality compared with weight-only quantization under the 4-bit setting. This indicates that the activations of video generation models are much harder to quantize than the weights. Similar observations have also been reported in previous studies (Zhao et al., 2025a; Tian et al., 2024).

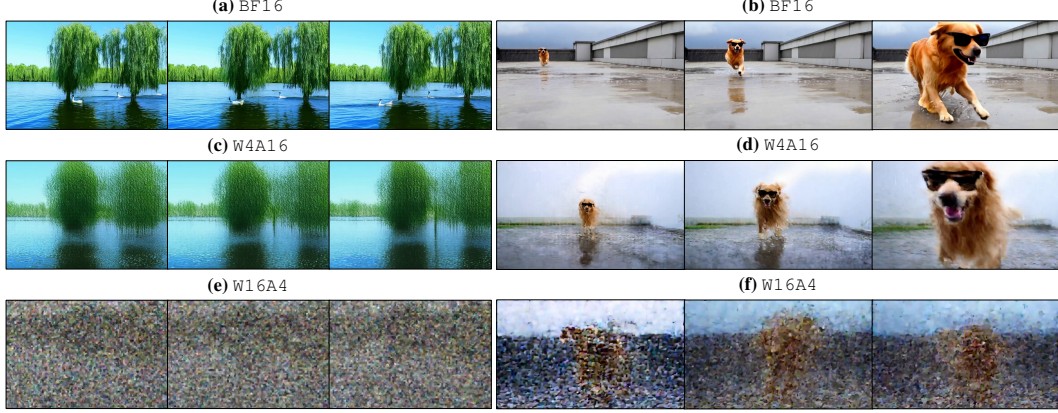

**(a)** `BF16`  **(b)** `BF16`

**(c)** `W4A16`  **(d)** `W4A16`

**(e)** `W16A4`  **(f)** `W16A4`

**Figure B:** Performacne for weight-only quantization *vs.* activation-only quantization. We employ Min-max (Nagel et al., 2021) *per-channel* weight quantization and *per-token* activation quantization for (*Left*) CogVideoX-2B (Yang et al., 2025) and (*Right*) `Wan` 1.3B (WanTeam et al., 2025).

## M Results for Image Generation

Our theoretical insight is broadly applicable to QAT. Additionally, the proposed QAT strategy is independent of model architecture and data type, so it can be transferred to other domains. However, further studies are needed to confirm whether the same strategy and insight can deliver similar significant improvements in other tasks. To be specific, the large gains we report for video generation

rely on the observed behavior of the gradient norm and the singular values during QAT for video diffusion. Here, we report the initial results for image generation in Tab. Q, which show that our approach can also achieve non-negligible performance enhancement without additional inference overhead. We will explore more about this in the future.

**Table Q:** `W4A4` quantization results for SD3-medium (Esser et al., 2024b). We evaluate FID (Heusel et al., 2018) and CLIP score (Hessel et al., 2022) on the MJHQ-30K (Li et al., 2024a) dataset and employ GenEval (Ghosh et al., 2023) to further measure text-image alignment.

| Method | FID↓ | CLIP Score↑ | GenEval↑ |
|---|---|---|---|
| Full Prec. | 11.92 | 27.83 | 0.62 |
| LSQ (Esser et al., 2020b) | 14.87 | 27.72 | 0.56 |
| EfficientDM (He et al., 2024) | 15.23 | 27.11 | **0.61** |
| Q-DM (Li et al., 2023b) | 13.82 | 27.68 | 0.59 |
| QVGen (Ours) | **12.24** | **27.85** | **0.61** |

# N   PROFILING AND PROJECTED GAINS FROM KERNEL FUSION

**Table R:** Latency breakdown (ms) and `INT4` GEMM throughput on A800. For ease of analysis, we adopt the shapes from `Wan 1.3B`. A DiT block for the model is composed of Self-Attention, Cross-Attention, and FFN. "`q1/k1/v1/o1`" and "`q2/k2/v2/o2`" denote projections in Self-Attention and Cross-Attention, respectively.

| Op. | $(M, N, K)$ | Quant | INT4GEMM | DeQuant | Total | Quant (%) | INT4GEMM (%) | DeQuant (%) | TOPS |
|---|---|---|---|---|---|---|---|---|---|
| `q1/k1/v1/o1/q2/o2` | $(32760, 1536, 1536)$ | 0.041 | 0.186 | 0.114 | 0.341 | 12.0 | 54.5 | 33.5 | 831.08 |
| `k2/v2` | $(512, 1536, 1536)$ | 0.001 | 0.007 | 0.002 | 0.010 | 11.2 | 72.3 | 16.5 | 345.13 |
| `up_proj` | $(32760, 8960, 1536)$ | 0.041 | 1.246 | 0.661 | 1.948 | 2.10 | 64.0 | 33.9 | 721.76 |
| `down_proj` | $(32760, 1536, 8960)$ | 0.232 | 1.031 | 0.117 | 1.380 | 16.8 | 74.7 | 8.48 | 872.27 |

To better understand the efficiency bottlenecks in our current non-fused `INT4` implementation, we profile representative transformer operators on an NVIDIA A800 (SM80) by annotating stages such as activation quantization (`Quant`), `INT4` general matrix multiplication (`INT4GEMM`), and dequantization (`DeQuant`) with NVTX ranges (`torch.cuda.nvtx.range_push/pop`). For each operator in Tab. R, we report its GEMM dimensions $(M, K, N)$ (*i.e.*, $[M, K] \times [K, N]$), per-stage latency, the fraction of the total operator time, and the effective `INT4` GEMM throughput computed as

$$\text{TOPS} = \frac{2MNK}{t_{\text{INT4GEMM}}} \div 10^{12}, \tag{U}$$

where $t_{\text{INT4GEMM}}$ is the measured `INT4GEMM` time in seconds. Across all tested shapes, the `INT4` GEMM kernels achieve 345–872 TOPS, within the same order of magnitude as the A800's `INT4` tensor-core peak (1248 TOPS). However, the surrounding non-GEMM stages (activation quantization and dequantization) still account for 25–45% of the operator latency, primarily due to extra global memory traffic and the absence of fused epilogues. Given the measured GEMM time fraction $G$ and assuming kernel fusion removes a fraction $r$ of non-GEMM overhead (*e.g.*, fusing quant/dequant and avoiding intermediate reads/writes), the achievable speedup is approximated by

$$\rho \approx \frac{1}{G + (1-G)(1-r)}. \tag{V}$$

Using our NVTX-derived $G$ values, $r = 0.6$ yields $\rho \approx 1.18\times$ (`down_proj`), $1.20\times$ (`k2/v2`), $1.28\times$ (`up_proj`), $1.38\times$ (`q1/k1/v1/o1/q2/o2`); and $r \approx 0.8$ yields $\rho \approx 1.25\times$, $1.28\times$, $1.40\times$, and $1.57\times$, respectively. These results indicate that a $1.2$–$1.6\times$ per-layer speedup is a realistic target once fusion is introduced.

# O   BREAKDOWN LATENCY ANALYSIS

Because a video DiT is implemented as a stack of identical blocks, we report the latency breakdown of a single DiT block to estimate its end-to-end impact (see Tabs. S-U). Within this block, attention computation (51.8%) is the dominant cost, while linear projections (24.5%) account for a large share of the remaining latency. To be noted, components other than linear projections can be accelerated by orthogonal strategies:

- *Attention*: Sparse attention, such as SVG (Xi et al., 2025), achieves a $1.73\times$ speedup for Self-Attention computation (31.48 *vs.* 18.23).

- *Other*: This category is largely composed of memory-bound operations, including `RoPE`, `norm`, and `reshape`, *etc*. These operations often launch many small kernels, so techniques such as CUDA Graphs and `torch.compile` can reduce dispatch overhead and enable more effective kernel fusion. Additionally, combined with fused and layout-aware kernels (Xi et al., 2025), the runtime of this category can be reduced by $5.59\times$ (14.64 *vs.* 2.620).

With these strategies applied, linear projections occupy a non-trivial $41.38\%$ of the block runtime. Therefore, reducing the latency of the linear projections is an important step toward further end-to-end speedups. In this work, `W4A4` quantization achieves a $2.52\times$ speedup for these linear projections (15.14 *vs.* 5.991). In addition, we plan to extend QVGen to `W4A4` attention quantization, which can further accelerate attention computation.

**Table S:** Latency breakdown (ms) for a DiT block (implemented in `torch`) on A800 (Wan 1.3B).

| Component | Time (ms) | Share (%) |
|---|---|---|
| Attention | 32.08 | 51.8 |
| Linear Projections | 15.14 | 24.5 |
| Other | 14.64 | 23.7 |

**Table T:** Latency breakdown (ms) for linear projections.

| Linear projections | Time (ms) |
|---|---|
| `q1/k1/v1/o1/q2/o2` | 0.977 |
| `k2/v2` | 0.038 |
| `up_proj` | 5.283 |
| `down_proj` | 3.913 |

**Table U:** Latency breakdown (ms) for attention.

| Attention | Time (ms) |
|---|---|
| Self-Attention | 31.48 |
| Cross-Attention | 0.597 |

## P  QUALITATIVE RESULTS

In this section, we present random samples generated by video DMs without cherry-picking, as exhibited from Figs. C-J. For a detailed comparison, **zoom in** to closely examine the relevant frames.

3**-bit quantization.** As shown in Figs. C and D, our method QVGen far outperforms other baselines under 3-bit quantization. Although 3-bit quantization still introduces noticeable performance degradation, especially for huge DMs (see Figs. E and F), we believe QVGen represents a promising step toward practical ultra-low-bit video DMs.

4**-bit quantization.** As depicted in Figs. G and H, previous QAT methods fail to deliver satisfactory results. In contrast, our method QVGen achieves video quality that closely approaches that of the full-precision model. Furthermore, for huge models (see Figs. I and J), QVGen consistently maintains high visual fidelity and effectively preserves generation quality.

## Q  THE USE OF LARGE LANGUAGE MODELS

We acknowledge the use of large language models (LLMs), such as OpenAI's GPT-5, as a writing-assistance tool in this work. Their role was strictly limited to proofreading and rephrasing sentences to enhance linguistic quality, without any contribution to the research ideation or experimental results.

**(a)** `BF16`                                  **(b)** `W3A3` QVGen (Ours)

**(c)** `W3A3` EfficientDM (He et al., 2024)    **(d)** `W3A3` Q-DM (Li et al., 2023b)

**(e)** `W3A3` LSQ (Esser et al., 2020b)

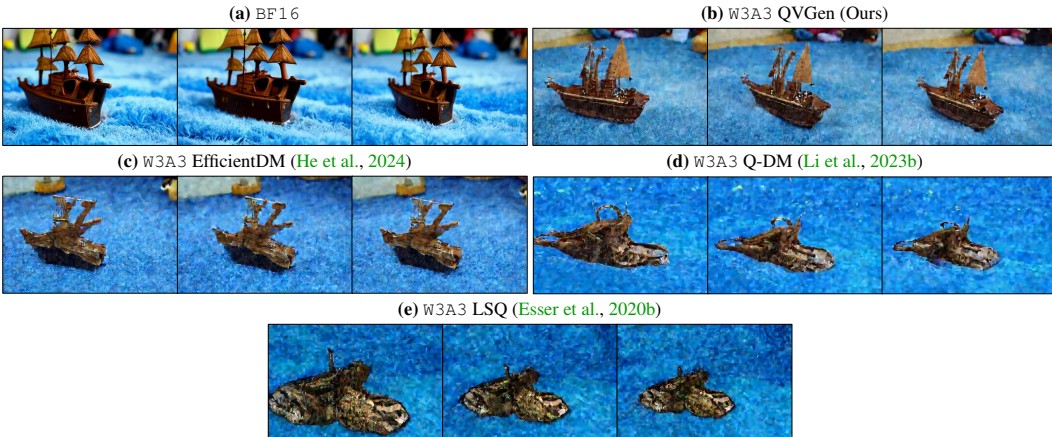

Text prompt: *"A detailed wooden toy ship with intricately carved masts and sails is seen gliding smoothly over a plush, blue carpet that mimics the waves of the sea. The ship's hull is painted a rich brown, with tiny windows. The carpet, soft and textured, provides a perfect backdrop, resembling an oceanic expanse. Surrounding the ship are various other toys and children's items, hinting at a playful environment. The scene captures the innocence and imagination of childhood, with the toy ship's journey symbolizing endless adventures in a whimsical, indoor setting."*

**Figure C:** Comparison of samples generated by full-precision and 3-bit CogVideoX-2B (Yang et al., 2025).

**(a)** `BF16`                                  **(b)** `W3A3` QVGen (Ours)

**(c)** `W3A3` EfficientDM (He et al., 2024)    **(d)** `W3A3` Q-DM (Li et al., 2023b)

**(e)** `W3A3` LSQ (Esser et al., 2020b)

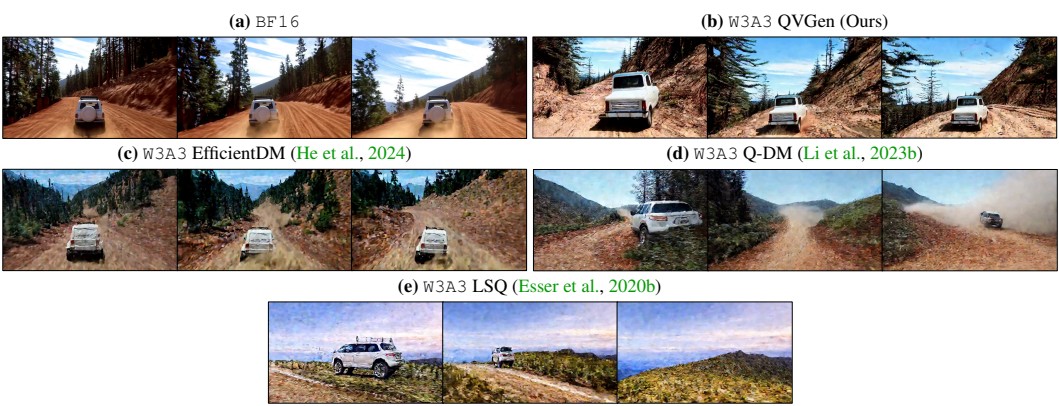

Text prompt: *"The camera follows behind a white vintage SUV with a black roof rack as it speeds up a steep dirt road surrounded by pine trees on a steep mountain slope, dust kicks up from its tires, the sunlight shines on the SUV as it speeds along the dirt road, casting a warm glow over the scene. The dirt road curves gently into the distance, with no other cars or vehicles in sight. The trees on either side of the road are redwoods, with patches of greenery scattered throughout. The car is seen from the rear following the curve with ease, making it seem as if it is on a rugged drive through the rugged terrain. The dirt road itself is surrounded by steep hills and mountains, with a clear blue sky above with wispy clouds."*

**Figure D:** Comparison of samples generated by full-precision and 3-bit `Wan` 1.3B (WanTeam et al., 2025).

**(a)** `BF16`

**(b)** `W3A3` QVGen (Ours)

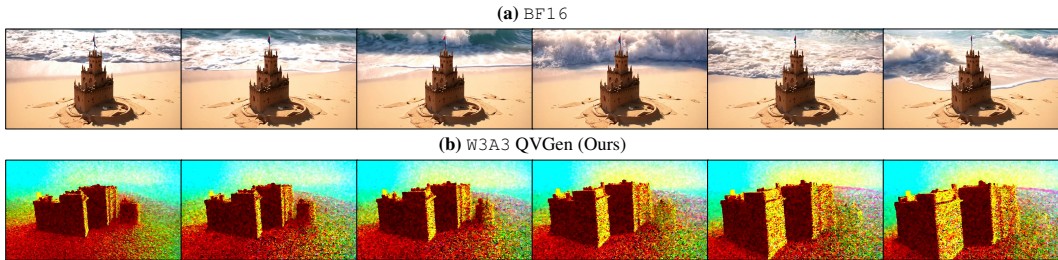

Text prompt: *"On a sunlit beach, a small, intricately detailed sandcastle stands near the shoreline, its turrets and walls casting delicate shadows on the golden sand. As the gentle waves lap nearby, the castle begins to transform, growing taller and more elaborate with each passing moment. Its towers stretch skyward, adorned with seashells and seaweed, while intricate patterns emerge on its expanding walls. The sun casts a warm glow, highlighting the castle's evolving grandeur. Finally, it stands as a majestic fortress, complete with a moat and flags fluttering in the breeze, a testament to the magic of imagination and the sea's timeless beauty."*

**Figure E:** Comparison of samples generated by full-precision and 3-bit CogVideoX1.5-5B (Yang et al., 2025).

**(a)** `BF16`

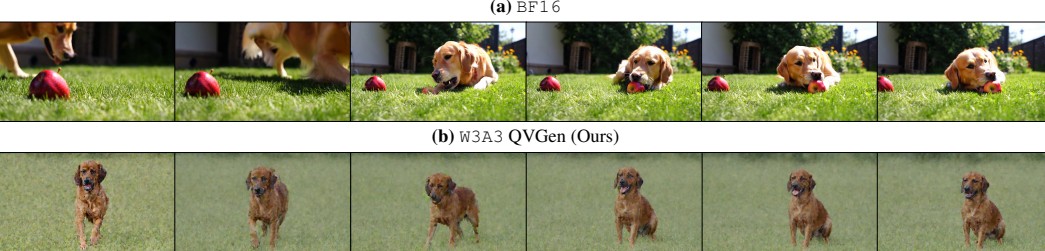

**(b)** `W3A3` QVGen (Ours)

Text prompt: *"A medium-sized golden retriever is initially positioned to the left of a juicy red apple. The dog, wagging its tail, notices something interesting and begins to run towards the apple, eventually coming to a playful stop right in front of it. The background is a sunny backyard with green grass and some flowers in the distance. The scene transitions smoothly, capturing the dog's curious and lively nature. The video is filmed in a dynamic style, with close-ups and medium shots to highlight the dog's movements and expressions. The lighting is bright and natural, emphasizing the textures of the dog's fur and the apple."*

**Figure F:** Comparison of samples generated by full-precision and 3-bit `Wan` 14B (WanTeam et al., 2025).

**(a)** `BF16`                **(b)** `W4A4` QVGen (Ours)

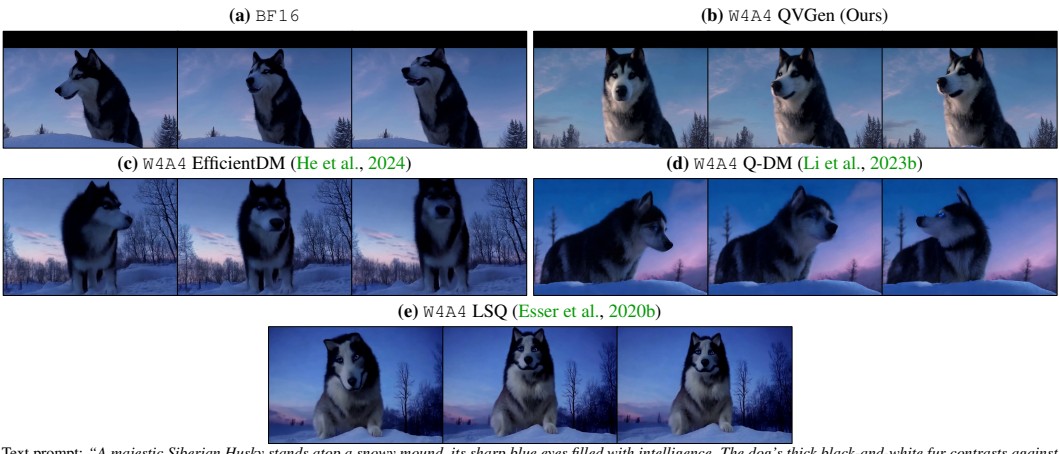

**(c)** `W4A4` EfficientDM (He et al., 2024)     **(d)** `W4A4` Q-DM (Li et al., 2023b)

**(e)** `W4A4` LSQ (Esser et al., 2020b)

Text prompt: *"A majestic Siberian Husky stands atop a snowy mound, its sharp blue eyes filled with intelligence. The dog's thick black-and-white fur contrasts against the soft twilight sky, where wisps of clouds drift peacefully. Bare trees surround the scene, standing as quiet guardians in the fading light."*

**(f)** `BF16`                **(g)** `W4A4` QVGen (Ours)

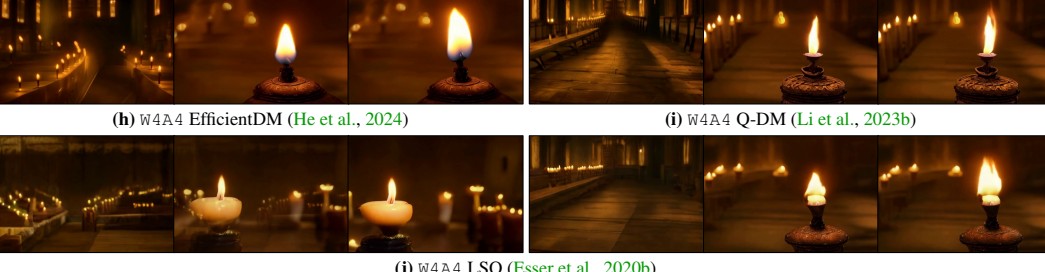

**(h)** `W4A4` EfficientDM (He et al., 2024)     **(i)** `W4A4` Q-DM (Li et al., 2023b)

**(j)** `W4A4` LSQ (Esser et al., 2020b)

Text prompt: *"In the heart of a grand, medieval hall, the scene is bathed in a warm, golden glow. A long wooden table stretches into the distance, adorned with flickering candles that cast a soft, inviting light. The air is filled with a sense of timelessness and reverence. A single, vibrant candle burns brightly in the foreground, its flame dancing gently atop a small, ornate holder."*

**Figure G:** Comparison of samples generated by full-precision and 4-bit CogVideoX-2B (Yang et al., 2025).

**(a)** BF16            **(b)** W4A4 QVGen (Ours)

**(c)** W4A4 EfficientDM (He et al., 2024)      **(d)** W4A4 Q-DM (Li et al., 2023b)

**(e)** W4A4 LSQ (Esser et al., 2020b)

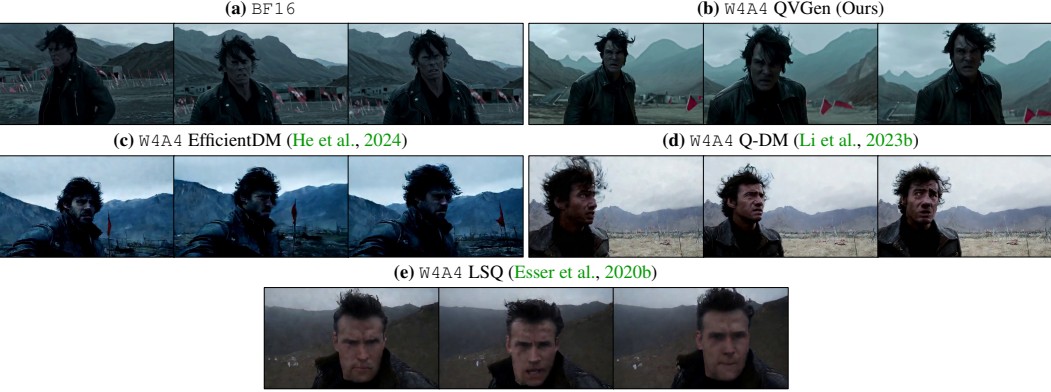

Text prompt: *"A man with tousled dark hair stands in a dramatic landscape, his eyes blazing with fury as he surveys the chaotic scene around him. Clad in a rugged leather jacket, he turns slightly, revealing a determined posture amid a backdrop of crumbling mountains and a valley littered with abandoned structures and scattered flags. The sky is overcast, adding a somber tone to the atmosphere, accentuating his emotional intensity. The camera captures a medium shot, focusing on his tense expression and the desolation surrounding him. The visual style is cinematic with high contrast, enhancing the grim and powerful mood of the moment."*

**(f)** BF16            **(g)** W4A4 QVGen (Ours)

**(h)** W4A4 EfficientDM (He et al., 2024)      **(i)** W4A4 Q-DM (Li et al., 2023b)

**(j)** W4A4 LSQ (Esser et al., 2020b)

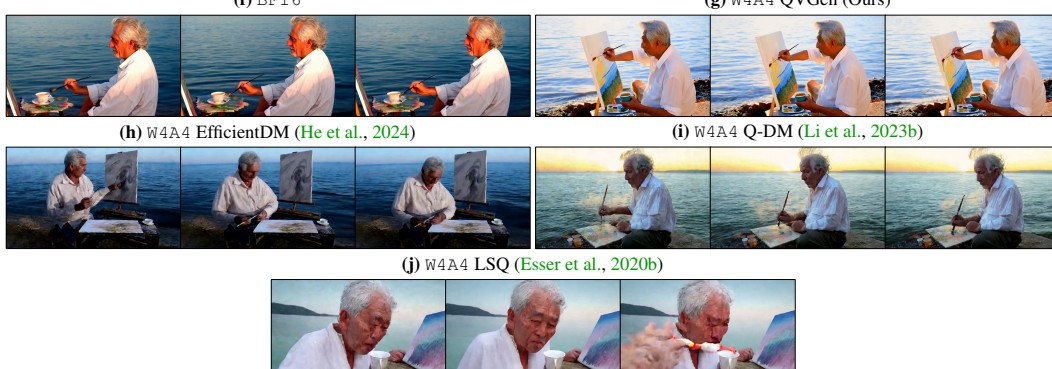

Text prompt: *"An elderly gentleman, with a serene expression, sits at the water's edge, a steaming cup of tea by his side. He is engrossed in his artwork, brush in hand, as he renders an oil painting on a canvas that's propped up against a small, weathered table. The sea breeze whispers through his silver hair, gently billowing his loose-fitting white shirt, while the salty air adds an intangible element to his masterpiece in progress. The scene is one of tranquility and inspiration, with the artist's canvas capturing the vibrant hues of the setting sun reflecting off the tranquil sea."*

**Figure H:** Comparison of samples generated by full-precision and 4-bit Wan 1.3B (WanTeam et al., 2025).

**(a)** BF16

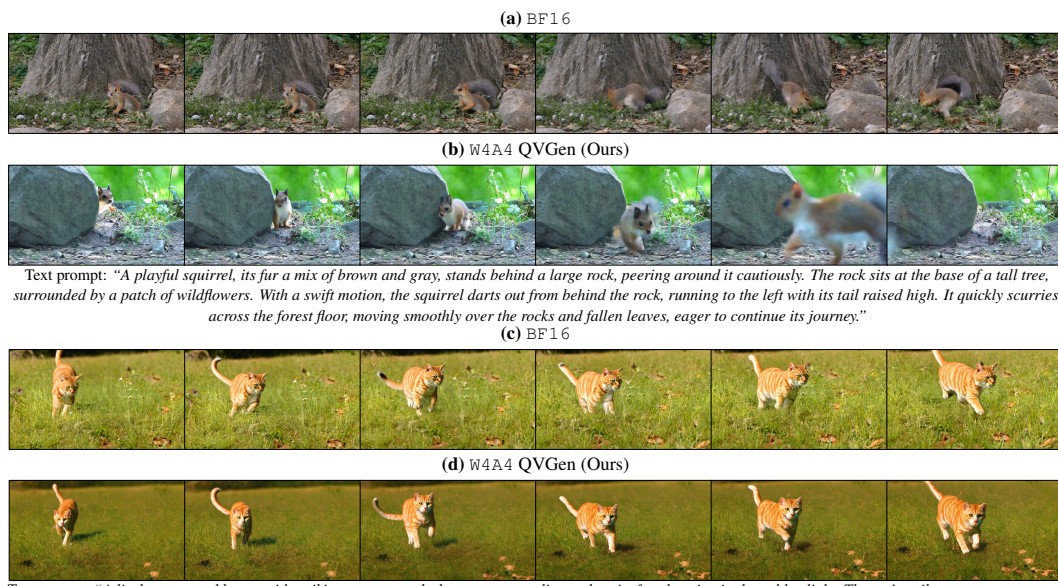

**(b)** W4A4 QVGen (Ours)

Text prompt: *"A playful squirrel, its fur a mix of brown and gray, stands behind a large rock, peering around it cautiously. The rock sits at the base of a tall tree, surrounded by a patch of wildflowers. With a swift motion, the squirrel darts out from behind the rock, running to the left with its tail raised high. It quickly scurries across the forest floor, moving smoothly over the rocks and fallen leaves, eager to continue its journey."*

**(c)** BF16

**(d)** W4A4 QVGen (Ours)

Text prompt: *"A lively orange tabby cat with striking green eyes dashes across a sunlit meadow, its fur gleaming in the golden light. The cat's agile movements create a blur of orange against the lush green grass, as it leaps over small wildflowers and navigates around scattered fallen leaves. Its tail flicks with excitement, and its ears are perked up, capturing every sound in the serene environment. The scene captures the essence of freedom and playfulness, with the cat's paws barely touching the ground, leaving a trail of gentle rustling in its wake."*

**Figure I:** Comparison of samples generated by full-precision and 4-bit CogVideoX1.5-5B (Yang et al., 2025).

**(a)** `BF16`

**(b)** `W4A4` QVGen (Ours)

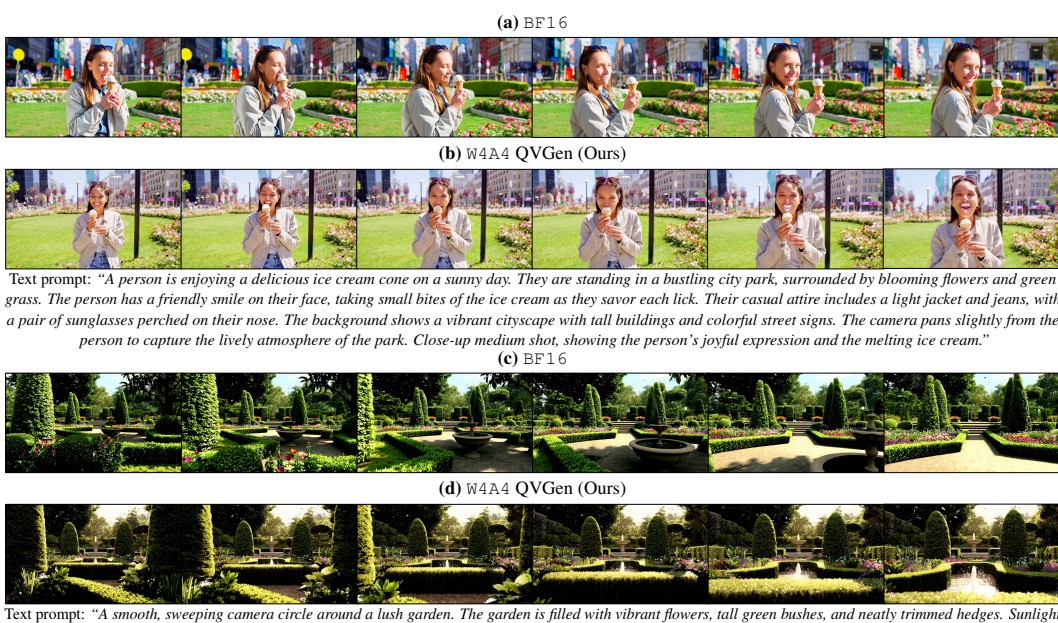

Text prompt: *"A person is enjoying a delicious ice cream cone on a sunny day. They are standing in a bustling city park, surrounded by blooming flowers and green grass. The person has a friendly smile on their face, taking small bites of the ice cream as they savor each lick. Their casual attire includes a light jacket and jeans, with a pair of sunglasses perched on their nose. The background shows a vibrant cityscape with tall buildings and colorful street signs. The camera pans slightly from the person to capture the lively atmosphere of the park. Close-up medium shot, showing the person's joyful expression and the melting ice cream."*

**(c)** `BF16`

**(d)** `W4A4` QVGen (Ours)

Text prompt: *"A smooth, sweeping camera circle around a lush garden. The garden is filled with vibrant flowers, tall green bushes, and neatly trimmed hedges. Sunlight filters through the leaves, casting dappled shadows on the ground. A small fountain sits in the center, gently spraying water into the air. Birds chirp and flutter among the branches. The camera gradually moves from a wide shot of the entire garden to closer views of individual plants and the intricate details of the landscape. The overall atmosphere is serene and inviting, with a soft, natural lighting style. Wide to medium shot, pans smoothly around the garden."*

**Figure J:** Comparison of samples generated by full-precision and 4-bit `Wan 14B` (WanTeam et al., 2025).

