# OpenReview forum: "QVGen: Pushing the Limit of Quantized Video Generative Models"
_ICLR.cc/2026/Conference — ICLR 2026 Poster_

### Official Review · Reviewer_uekc · 2025-10-30

**Soundness:** 3
**Presentation:** 3
**Contribution:** 3
**Rating:** 6
**Confidence:** 3

**Summary:**

This paper introduces QVGen, a novel Quantization-Aware Training (QAT) framework designed to enable high-quality video generation under extremely low-bit (3/4-bit) quantization, a task where previous methods fail. The key innovation is a two-stage process: first, it stabilizes training and improves convergence by adding lightweight auxiliary modules to mitigate quantization error, which is theoretically and empirically shown to reduce the gradient norm; second, it progressively eliminates these modules during training via a "rank-decay" strategy that uses Singular Value Decomposition (SVD) and a rank-based regularization to identify and shrink low-impact components to zero, resulting in a final quantized model with no inference overhead.

**Strengths:**

1. Well-Motivated Contribution: This paper addresses the challenge of ultra-low-bit QAT for large-scale video diffusion models, filling a significant gap in the literature which has primarily focused on image models.

2. Strong Theoretical and Empirical Foundation: The paper provides a solid theoretical analysis linking gradient norm reduction to improved QAT convergence, and then designs a method (the auxiliary module Φ) specifically to achieve this. The empirical results consistently show lower gradient norms and training loss.

3. Impressive and Extensive Experiments: The evaluation is comprehensive, testing on four state-of-the-art models ranging from 1.3B to 14B parameters. The results are compelling, showing that QVGen is the first method to achieve full-precision comparable quality with 4-bit quantization and significantly outperforms all baselines in 3-bit settings.

4. Practical and Efficient Solution: The proposed "rank-decay" strategy is a clever way to gain the training benefits of the auxiliary modules without incurring any inference cost, making the final model directly deployable with standard low-bit kernels. The reported ~4x memory reduction and up to 1.7x speedup are substantial.

**Weaknesses:**

1. Computational Cost of QAT: While the final model is efficient, the QAT process itself is expensive, involving iterative SVD operations and training on up to 32 H100 GPUs for large models. The paper does not deeply discuss the trade-offs between this training cost and the resulting inference savings.

2. Limited Analysis of "Rank-Decay": The strategy is shown to work, but the analysis of why the singular values of W_Φ evolve to have an increasing number of small components is somewhat surface-level. A deeper investigation into the dynamics between the quantized model and the auxiliary module during training would be valuable.

3. Ablation on Simpler Alternatives: The paper compares against other fine-grained decay strategies (Sparse, Residual Quantization), but it would be strengthened by also ablating against a simple scheduled decay (e.g., linearly reducing the magnitude of all parameters in Φ to zero) to more clearly isolate the benefit of the SVD-based, rank-aware approach.

**Questions:**

Hyperparameter Sensitivity: How sensitive is the final performance to the key hyperparameters, such as the initial rank r=32 and the shrinking ratio λ=1/2? Was there a systematic process for selecting these values across different model architectures and sizes?

---

> ### Author Response · Authors · 2025-11-19
> **Response to Reviewer uekc**
>
> Thanks to the reviewer for the constructive comments.
>
> - **Computational cost of QAT.** We have provided an analysis of computational costs in Tab. A and supplement a more detailed discussion in Sec. 4.4. Here, we further clarify the trade-off between these training costs and the resulting inference savings. Our method trains the 2B-parameter model on 8 H100 GPUs in less than 1.5 days and the 14B-parameter model on 32 H100 GPUs in fewer than 6 days, yielding $\sim4\times$ memory savings with $1.21\times$ / $1.44\times$ practical speedups (or $1.70\times$ / $2.63\times$ when combined with SVG). We attribute the modest speedup mainly to the lack of optimized CUDA kernels, which is not the primary focus of this work. As discussed in our response to the acceleration gains [Reviewer eVaC], these kernels could achieve more than a $1.5×$ improvement over the current implementation with further optimization. In future work, we plan to implement faster CUDA kernels and investigate more efficient training techniques for QVGen, while preserving its strong performance.
> - **Analysis of *rank-decay*.** Because of our focus on improving QAT performance, we offer a potential justification for the “rank-decay” behavior and will continue to explore more about this. Regarding that $\mathbf{W}_{\Phi}$ contains many very small singular values, we attribute this to the low‑rank nature of the quantization error in video diffusion. Similar findings have been reported for LLM quantization [a] and image diffusion models quantization [b]. For the behavior that the small singular values become increasingly pronounced as QAT proceeds, we posit that the main weight $\mathbf{W}$ gradually learns to absorb quantization errors. Consequently, $\mathbf{W}\_{\Phi}$ collapses to an even lower rank and plays an ever‑diminishing role in error compensation. Besides these analyses, a deeper investigation into the dynamics between the quantized model and the auxiliary module is an interesting direction for future work.
> - **Simple scheduled decay.** Thanks for the helpful suggestion. In the following table, we investigate the suggested simple scheduled decay for $\Phi$ (*i.e.*, linearly shrinking all parameters in $\Phi$ to zero) to better highlight the advantage of our rank-decay strategy. The corresponding results are also added to Tab. 6.
>
>     *We employ Wan 1.3B with W4A4 quantization here.*
>
>     | Method | Imaging Quality$\uparrow$ | Aesthetic Quality$\uparrow$ | Dynamic Degree$\uparrow$ | Scene Consistency$\uparrow$ | Overall Consistency$\uparrow$ |
>     | --- | --- | --- | --- | --- | --- |
>     | linear-decay | 60.82 | 52.81 | 73.19 | 13.34 | 21.87 |
>     | rank-decay | 63.08 | 54.67 | 77.78 | 15.32 | 23.01 |
> - **Hyperparameter sensitivity.**  In practice, we observe that the final performance is insensitive to the chosen hyperparameters $r=32$ and $\lambda=\frac{1}{2}$ across settings. We use these values for all experiments in the paper (except Tabs. 4 and 5, which ablate $r$ and $\lambda$, respectively), and our approach consistently outperforms the baselines (*e.g.*, Tabs. 1, 2, and H). To further improve performance with a more systematic procedure for setting these hyperparameters, one possible scheme is as follows: before QAT, run a grid search over $r$ to find the smallest value that keeps the model’s output error within a given bound. Then, during QAT, perform a similar grid search for $\lambda$  at the start of each decay phase to select a value that also maintains the output error within the desired range. A full study is left for future work.
>
> [a] Zhang, Cheng et al. “LQER: Low-Rank Quantization Error Reconstruction for LLMs.” *ArXiv* abs/2402.02446 (2024): n. pag.
>
> [b] Li, Muyang et al. “SVDQuant: Absorbing Outliers by Low-Rank Components for 4-Bit Diffusion Models.” *ArXiv* abs/2411.05007 (2024): n. pag.

---

> ### Comment · Reviewer_uekc · 2025-11-28
>
> Thank you for the comprehensive rebuttal. With my concerns satisfactorily addressed and the paper's valuable insights in mind, I am raising my score to 8.

---

> > ### Author Response · Authors · 2025-11-28
> > **Tnanks for your feedback**
> >
> > Dear Reviewer uekc
> >
> > Thank you for your feedback! We appreciate the constructive reviews for improving our work.
> >
> > Best regards,
> >
> > Authors of Paper #11845

---

### Official Review · Reviewer_eVaC · 2025-11-01

**Soundness:** 3
**Presentation:** 3
**Contribution:** 3
**Rating:** 6
**Confidence:** 4

**Summary:**

The paper introduces QVGen, a quantization-aware training (QAT) method designed to enable high-quality video diffusion models under extremely low-bit quantization (4-bit or 3-bit). The paper begins with a theoretical analysis showing that reducing the gradient norm is key to stabilizing and improving convergence in training. Based on this insight, the paper introduces auxiliary modules $\Phi$ during QAT to compensate for weight quantization errors and smooth optimization. To avoid extra inference overhead from these modules, the paper proposes a rank-decay mechanism, progressively eliminating $\Phi$ by decomposing its weights via singular value decomposition (SVD) and decaying low-contributing components using a rank-based regularization schedule. Extensive experiments on state-of-the-art models demonstrate that QVGen achieves full-precision comparable performance in 4-bit settings and sets new records under 3-bit quantization, outperforming existing QAT and PTQ baselines such as Q-DM, EfficientDM, and SVDQuant. The method also improves memory efficiency and inference speed while maintaining compatibility with standard low-bit kernels.

**Strengths:**

1. **Well-motivated methodology**: The proposed approach is grounded in a clear motivation. The authors observe that the auxiliary module $\Phi$ exhibits rank decay during quantization-aware training and effectively leverage this property through their rank-decay mechanism. This insight is both intuitive and novel.
2. **Comprehensive experimental validation**: The paper provides extensive experimental results across multiple large-scale video diffusion models, supported by detailed ablation studies and qualitative visual comparisons.
3. **Strong empirical performance**: QVGen achieves consistently superior results compared to existing QAT and PTQ baselines, showing that the proposed framework is highly effective in maintaining video generation quality under ultra-low-bit quantization while improving efficiency.

**Weaknesses:**

1. **Theory disconnected from the main method**: The theoretical analysis in the early part of the paper appears largely disconnected from the core contributions. Although the authors claim that the auxiliary module $\Phi$ is motivated by this theory, the linkage is tenuous, and the theoretical result itself is rather weak. As a result, the theory does not substantially contribute to the understanding or justification of the proposed framework (although I think it is totally fine to be motivated empirically).

2. **Limited acceleration gains**: The acceleration gains reported by QVGen are not particularly large, which somewhat limits its practical impact on efficiency. However, the authors explicitly acknowledge this limitation and attribute it to the absence of kernel fusion optimizations, which is reasonable.

**Questions:**

1. The proposed QVGen framework focuses on compensating quantization errors in the weights through the auxiliary module $\Phi$. However, the paper does not analyze the effect of activation quantization separately. Could the authors provide results or discussion on how the model performs when only activations are quantized to low-bit precision while keeping the weights in full precision? This would help clarify whether the main difficulty in quantizing video diffusion models arises more from weights or activations.

2. The paper mentions that the current acceleration is limited because kernel fusion is not applied. To better understand the efficiency aspect, could the authors provide additional profiling results—such as the achieved TFLOPs of the INT4 GEMM kernel used—and an end-to-end inference time breakdown? It would be very helpful if the authors could use a profiling tool (e.g., nsys profile) with `torch.cuda.nvtx.range_push` and `torch.cuda.nvtx.range_pop` tags to visualize where most time is spent and estimate how much of the overhead could be mitigated by kernel fusion, even without implementing it. The author is encouraged to provide a table that describe how much time the qkv projection, o projection, ffn up projection, ffn down projection, self attention, and all other memory bound modules take in a single dit block.

---

> ### Author Response · Authors · 2025-11-19
> **Response to Reviewer eVaC (Part 1)**
>
> Thanks to the reviewer for the constructive comments.
>
> - **Connection between theory and the main method.**  We want to emphasize that the purpose of the theory connects QAT convergence with the gradient norm: a smaller gradient norm can lead to better QAT convergence. Although this is not the main contribution of the paper, it provides key insight for designing our method. Specifically, as a relatively small gradient norm is often associated with stable training dynamics [a], we introduce $\Phi$ to compensate for the aggressive quantization error (*i.e.*, mitigate loss spikes) and thus stabilize the QAT process. Our experimental results (see Sec. 3.1) confirm that the proposed method achieves relatively low gradient norms and thereby improves QAT performance.
> - **Effect of activation quantization.** First, we explain that the auxiliary module $\Phi$, although initialized from the weight quantization error, is to mitigate the aggressive output error of the quantized model during QAT for stable training. Thus, it considers both weight and activation quantization error during QAT. Additionally, following the reviewer’s suggestion, we examine the main difficulty in quantizing video DMs. Sec. M.6 (added in this rebuttal) visualizes the effects of weight-only and activation-only quantization on Wan 1.3B and CogVideoX-2B. Activation-only quantization leads to much stronger degradation in video quality than weight-only quantization, indicating that activations are substantially harder to quantize than weights, consistent with prior findings [b-c].
> - **Acceleration gains.** Thanks to the reviewer for the helpful suggestion. We profile our non‑fused INT4 pipeline on an NVIDIA A800 by annotating stages such as activation quantization, INT4 GEMM, and dequantization with NVTX ranges (`torch.cuda.nvtx.range_push/pop`), and we aggregate the nsys timelines to obtain per‑stage latencies. As shown in the table below, INT4 GEMM kernels sustain 345–872 effective TOPS across representative transformer shapes, which is within the same order of magnitude as the A800 INT4 tensor-core peak (1248 TOPS). However, non‑GEMM stages still account for 25–45% of the total operator latency, mainly due to additional global memory traffic and the absence of fused epilogues. Based on the measured GEMM fraction $G$, if kernel fusion removes a fraction $r$ of non‑GEMM overhead, the achievable speedup is $\rho \approx 1 / (G + (1−G)(1−r))$. Applying this formula, moderate fusion (*e.g.*, $r = 0.6$) yields about $1.2–1.4\times$ speedup, and stronger fusion (*e.g.*, $r \approx 0.8$) reaches about $1.3–1.6×$. This suggests that a $1.2–1.6×$ per‑layer improvement is realistically attainable once fusion is integrated. We plan to incorporate these fusion strategies in future work.  The above analyses can be found in Sec. O.
>
>     *Latency breakdown (ms) and INT4 GEMM ($[M, K] \times [K, N]$) throughput on A800 (Wan 1.3B). A DiT block is composed of Self‑Attention, Cross‑Attention, and FFN. “q1/k1/v1/o1” and “q2/k2/v2/o2” denote projections in Self‑Attention and Cross‑Attention, respectively.*
>
>     | Operator | M | K | N | quant (ms) | int4_gemm (ms) | dequant (ms) | total (ms) | quant (%) | int4_gemm (%) | dequant (%) | int4_gemm TOPS |
>     | --- | --- | --- | --- | --- | --- | --- | --- | --- | --- | --- | --- |
>     | q1/k1/v1/o1/q2/o2 | 32760 | 1536 | 1536 | 0.041 | 0.186 | 0.114 | 0.341 | 12.0 | 54.5 | 33.5 | 831.08 |
>     | k2/v2 | 512 | 1536 | 1536 | 0.001 | 0.007 | 0.002 | 0.010 | 11.2 | 72.3 | 16.5 | 345.13 |
>     | FFN up | 32760 | 1536 | 8960 | 0.041 | 1.246 | 0.661 | 1.948 | 2.10 | 64.0 | 33.9 | 721.76 |
>     | FFN down | 32760 | 8960 | 1536 | 0.232 | 1.031 | 0.117 | 1.380 | 16.8 | 74.7 | 8.48 | 872.27 |

---

> ### Author Response · Authors · 2025-11-19
> **Response to Reviewer eVaC (Part 2)**
>
> Additionally, because a video DiT consists of a stack of identical blocks, we report the latency breakdown of a single DiT block in the following tables to estimate its end-to-end impact. Within this block, attention computation (51.8%) is the dominant cost, while linear projections (24.5%) account for a large share of the remaining latency. To be noted, components other than linear projections can be accelerated by orthogonal strategies:
>
> - *Attention*: Sparse attention, such as SVG [d], achieves a $1.73\times$ speedup for Self-Attention computation (31.48 *vs.* 18.23).
> - *Other*: This category is largely composed of memory-bound operations, including `RoPE`, `norm`, and `reshape`, *etc*. These operations often launch many small kernels, so techniques such as CUDA Graphs and `torch.compile` can reduce dispatch overhead and enable more effective kernel fusion. Additionally, combined with fused and layout-aware kernels [d], the runtime of this category can be reduced by $5.59\times$ (14.64 *vs.* 2.620).
>
> With these strategies applied, linear projections occupy a non-trivial 41.38% of the block runtime. Therefore, reducing the latency of the linear projections is an important step toward further end-to-end speedups. In this work, W4A4 quantization achieves a $2.52\times$ speedup for these linear projections (15.14 *vs.* 5.991). Besides that, we plan to extend QVGen to W4A4 attention quantization, which can further accelerate attention computation. These analyses are included in Sec. P.
>
> *Latency breakdown (ms) for a DiT block (implemented in torch) on A800 (Wan 1.3B).*
>
> | Component | Time (ms) | Share (%) |
> | --- | --- | --- |
> | attention | 32.08 | 51.8 |
> | linear projections | 15.14 | 24.5 |
> | other (activation/norm/…) | 14.64 | 23.7 |
>
> *Latency breakdown (ms) for linear projections.*
>
> | Linear projections | Time (ms) |
> | --- | --- |
> | q1/k1/v1/o1/q2/o2 | 0.977 |
> | k2/v2 | 0.038 |
> | FFN up | 5.283 |
> | FFN down | 3.913 |
>
> *Latency breakdown (ms) for attention.*
>
> | Attention | Time (ms) |
> | --- | --- |
> | Self-Attention | 31.48 |
> | Cross-Attention | 0.597 |
>
> [a] Xie, Zeke et al. “On the Overlooked Pitfalls of Weight Decay and How to Mitigate Them: A Gradient-Norm Perspective.” *Neural Information Processing Systems* (2020).
>
> [b] Tian S, Chen H, Lv C, et al. Qvd: Post-training quantization for video diffusion models[C]//Proceedings of the 32nd ACM International Conference on Multimedia. 2024: 10572-10581.
>
> [c] Zhao, Tianchen et al. “ViDiT-Q: Efficient and Accurate Quantization of Diffusion Transformers for Image and Video Generation.” *ArXiv* abs/2406.02540 (2024): n. pag.
>
> [d] Xi, Haocheng et al. “Sparse VideoGen: Accelerating Video Diffusion Transformers with Spatial-Temporal Sparsity.” *ArXiv* abs/2502.01776 (2025): n. pag.

---

> > ### Comment · Reviewer_eVaC · 2025-11-20
> >
> > I appreciate the effort of the author. My concern is largely resolved. I will keep my score and I lean towards acceptance.

---

> > > ### Author Response · Authors · 2025-11-20
> > > **Thanks for your feedback**
> > >
> > > Dear Reviewer eVaC
> > >
> > > Thank you for your feedback! We appreciate the constructive reviews for improving our work.
> > >
> > > Best regards,
> > >
> > > Authors of Paper #11845

---

### Official Review · Reviewer_FHRA · 2025-11-01

**Soundness:** 3
**Presentation:** 3
**Contribution:** 3
**Rating:** 8
**Confidence:** 3

**Summary:**

This paper introduces QVGen, a novel Quantization-Aware Training (QAT) framework designed specifically for very low-bit (≤4-bit) video quantization. The core of this method is the introduction of an auxiliary module ($\Phi$) to stabilize the QAT process. To eliminate the inference overhead from this auxiliary module, the authors propose a "rank-decay" strategy, which uses Singular Value Decomposition (SVD) and rank regularization to progressively remove these modules during training. Experiments on video DMs ranging from 1.3B to 14B parameters show that QVGen achieves quality comparable to full-precision at 4-bit settings.

**Strengths:**

1. The first full QAT method for video generation models I'm aware of.

2. Experiments are conducted on four SOTA open-source video DMs (CogVideoX and Wan), with parameter scales from 1.3B to 14B, providing broad coverage.

3. It validates practical efficiency gains and demonstrates orthogonality with other acceleration techniques like SVG.

4. The provided experimental materials are comprehensive, and the ablation studies are extensive.

**Weaknesses:**

1. Since some other QAT methods are trained using only LoRA, a comparison of training time and memory (GPU VRAM) requirements against these methods should be provided for a comprehensive assessment of algorithm efficiency.

2. Quantization-related initialization settings should be specified, such as the choice of quantizer (e.g., granularity, symmetric/asymmetric) and which layers, if any, are not quantized.

3. In Fig.3, why the inital training loss of the proposed method is bettern than Q-DM? Did the proposed method use a better initialization strategy? Since the authors reproduce other QAT baselines, a detailed training loss curve comparison would greatly enchance the soundness of the paper.

**Questions:**

Please see the weaknesses above, and:

1. Can the proposed method be combined with SVDQuant? If so, could this combination be trained efficiently by updating only LoRA parameters?

---

> ### Author Response · Authors · 2025-11-19
> **Response to Reviewer FHRA**
>
> Thanks to the reviewer for the constructive comments.
>
> - **Training time and memory.** We conduct a comparative training efficiency analysis across different models and QAT baselines, as demonstrated in the following table. LSQ does not use distillation-based QAT (*i.e.*, no teacher forward pass) and is therefore the fastest to train. EfficientDM updates only LoRA parameters, which substantially reduces optimizer-state memory on GPUs. We believe such a strategy in EfficientDM can be combined with our method to further improve training efficiency, and we plan to explore this in follow-up studies. Relative to distillation-based QAT (*i.e.*, Q-DM), our method with low-rank $\Phi$ adds only $\sim1.02\times$ GPU-days and $\sim1.01\times$ peak GPU memory for Wan 1.3B model. To be noted, *all* baselines greatly fall short of our method in final performance, as illustrated in Sec. 4.2. In future work, we will delve into improving QVGen's training efficiency while maintaining its strong performance.
>
>     *Training costs across different methods and models on $H100$ GPUs.*
>
>     |  | Train. Time (GPU days)$\downarrow$ | Train. Mem. (GB/GPU)$\downarrow$ |
>     | --- | --- | --- |
>     | CogVideoX-2B |  |  |
>     | LSQ | 8.64 | 62.78 |
>     | Q-DM | 9.30 | 67.26 |
>     | EfficientDM | 8.97 | 44.27 |
>     | QVGen | 9.44 | 67.93 |
>     | Wan 1.3B |  |  |
>     | LSQ | 9.92 | 63.04 |
>     | Q-DM | 10.92 | 66.15 |
>     | EfficientDM | 10.68 | 42.74 |
>     | QVGen | 11.11 | 66.67 |
> - **Quantization settings.** We quantize all linear layers with asymmetric uniform quantization. Channel-wise static quantization is applied to weights, and token-wise dynamic quantization is applied to activations. This schema has already been given in Lines 140-141 and 302-304.
> - **Training loss.** The proposed method uses $\mathbf{W}-\mathrm{Q}\_b(\mathbf{W})$ to initialize $\mathbf{W}\_\Phi$  (Lines 216–217), which reduces quantization error. As a result, the initial training loss is lower than that of Q-DM. For the quantized DM *w/o* $\mathbf{W}_\Phi$, we use the same initialization as Q-DM: MSE calibration [a] for statically quantized weights; Min-Max calibration [a] for dynamically quantized activations. Moreover, we supplement a loss curve comparison across methods and models in Sec. H.
> - **Combination with SVDQuant.** Yes, our method can be combined with SVDQuant. In the following table, we first apply SVDQuant to obtain a weight-modified DM, the quantization parameters, and the LoRA matrices, which we reuse as $\Phi$. We then run QVGen, which progressively removes $\Phi$. This combination yields further gains, likely due to the strong initialization from SVDQuant. We also evaluate an option that updates only the LoRA parameters (*i.e.*, $\Phi$) and quantization parameters within this combination. It achieves sizable improvements over SVDQuant alone, but it still falls short of QVGen when model weights are updated. This suggests that QAT, which trains model weights, is currently important for video generation quantization. We will continue to explore more efficient ways to quantize video DMs while preserving performance. We add these results in Sec. J.
>
>     *W4A4 results of the combination with SVDQuant for Wan 1.3B. “^” denotes we freeze the weights of the DM and only finetune the quantization parameters and the introduced $\Phi$. “\*” means we employ a more fine-grained and performance-friendly quantization setting as in SVDQuant's paper (details can be found in Sec. E).*
>
>     | Method | Imaging Quality$\uparrow$ | Aesthetic Quality$\uparrow$ | Dynamic Degree$\uparrow$ | Scene Consistency$\uparrow$ | Overall Consistency$\uparrow$ |
>     | --- | --- | --- | --- | --- | --- |
>     | Full Prec. | 64.30 | 58.21 | 70.28 | 28.05 | 24.67 |
>     | SVDQuant* | 57.57 | 46.30 | 72.22 | 12.73 | 21.91 |
>     | QVGen | 63.08 | 54.67 | 77.78 | 15.32 | 23.01 |
>     | QVGen w/ SVDQuant | 63.64 | 56.23 | 77.42 | 17.65 | 23.89 |
>     | QVGen^ w/ SVDQuant | 61.38 | 52.76 | 75.85 | 14.12 | 22.47 |
>
> [a] Nagel, Markus, Marios Fournarakis, Rana A. Amjad, Yelysei Bondarenko, Mart Van Baalen, and Tijmen Blankevoort. "A White Paper on Neural Network Quantization." *ArXiv*, (2021). Accessed November 14, 2025. https://arxiv.org/abs/2106.08295.

---

> > ### Comment · Reviewer_FHRA · 2025-11-28
> > **response to rebuttal**
> >
> > Thank you for the supplementary experiments. I will maintain my clear acceptance (8).

---

> > > ### Author Response · Authors · 2025-11-28
> > > **Thanks for your feedback**
> > >
> > > Dear Reviewer FHRA
> > >
> > > Thank you for your feedback! We appreciate the constructive reviews for improving our work.
> > >
> > > Best regards,
> > >
> > > Authors of Paper #11845

---

### Official Review · Reviewer_RsTT · 2025-11-01

**Soundness:** 4
**Presentation:** 4
**Contribution:** 3
**Rating:** 8
**Confidence:** 3

**Summary:**

This paper proposes a quantization method for video diffusion transformers. The key idea is to introduce additional modules containing full-precision lora parameters to mitigate aggressive training losses, whose effectiveness of reducing gradient norm and quantization error has been demonstrated by the authors. Then, the authors devise a rank shrinking strategy, which reduces the rank to 0 to avoid additional stroage of full-precision weights. Extensive experiments demonstrate the effectiveness.

**Strengths:**

1. The work could be highly impactful for the community of quantized video generation models due to its state-of-the-art performance.
2. The analysis of the importance of reducing the gradient norm is valid and motivating for the proposed method.
3. Although the method first introduces full-precision parameters, the authors devise effective solutions to reduce the rank to even 0, which means eliminating the need for additional full-precision storage. From the results, such a two-stage pipeline is highly effective and superior to one-stage methods.
4. The experiments are very extensive and sufficient to demonstrate the effectiveness of quantizing video diffusion models.

**Weaknesses:**

I don't find so many weaknesses, but would like to list some minor points below:
1. It seems that the method is not tailored for video diffusion models and has potential for other models, like image generation and image backbone. The authors are encouraged to conduct experiments on these widely adopted benchmarks.
2. It is encouraged to include another baseline of fine-tuning the model using the same data under full precision, which is useful to reflect the effect introduced by additional data and fine-tuning.

**Questions:**

* Is it possible to directly regulate gradient norm by gradient normalization, clip, or direct optimization? These methods may yield inferior results, but can they outperform the original baseline introduced in Sec. 2?
Please refer to the weaknesses part above for other questions.

---

> ### Author Response · Authors · 2025-11-19
> **Response to Reviewer RsTT**
>
> Thanks to the reviewer for the constructive comments.
>
> - **Image generation.** It is more challenging to quantize video DMs than image DMs (see Sec. 1), so we focus on QAT for video DMs in this work. Even so, our method can also be used for other models. In the appendix, we have already reported initial results on image generation in Tab. Q, and in the rebuttal, we further include two baselines (LSQ and EfficientDM). As shown below, QVGen still outperforms these baselines across benchmarks.
>
>     *W4A4 Quantization results for SD3-medium [a]. We train SD3-medium for 2K steps with 16K images from the LAION-5B dataset [b] on 8 H100 GPUs. We evaluate FID and Clip score on the MJHQ-30K [c] dataset and employ GenEval [d] to further measure text-image alignment.*
>
>     |  | FID$\downarrow$ | CLIP$\uparrow$ | GenEval$\uparrow$ |
>     | --- | --- | --- | --- |
>     | Full Prec. | 11.92 | 27.83 | 0.62 |
>     | LSQ | 14.87 | 27.72 | 0.56 |
>     | EfficientDM | 15.23 | 27.11 | 0.61 |
>     | Q-DM | 13.82 | 27.68 | 0.59 |
>     | QVGen (Ours) | 12.24 | 27.85 | 0.61 |
> - **Baseline of fine-tuning.** Thanks for the helpful suggestion. We add full fine-tuning results in the table below and in Tab. 1 of the paper. The results show that, when we fine-tune the full-precision model on the same data used for QVGen, performance improves on some metrics, but drops on others (*e.g.*, Scene Consistency and Background Consistency). We suspect this is because the original full-precision model was trained on carefully curated data, as suggested by Fig. 3 in Wan’s report, whereas our additional data preprocessing is relatively lightweight. Despite this, for low-bit quantized models, we observe that their performance can be largely recovered without relying on very high-quality data in the paper. However, using higher-quality data may further improve quantization performance, and we plan to explore this in future work.
>
>
>     | Method | Imaging Quality$\uparrow$ | Aesthetic Quality$\uparrow$ | Motion Smoothness$\uparrow$ | Dynamic Degree$\uparrow$ | Background Consistency$\uparrow$ | Subject Consistency$\uparrow$ | Scene Consistency$\uparrow$ | Overall Consistency$\uparrow$ |
>     | --- | --- | --- | --- | --- | --- | --- | --- | --- |
>     | CogVideoX-2B |  |  |  |  |  |  |  |  |
>     | Full Prec. | 59.15 | 54.49 | 97.43 | 67.78 | 94.79 | 92.82 | 36.24 | 25.06 |
>     | Full Finetuning | 61.34 | 56.53 | 98.59 | 65.39 | 93.84 | 93.43 | 34.99 | 25.50 |
>     | Wan 1.3B |  |  |  |  |  |  |  |  |
>     | Full Prec. | 64.30 | 58.21 | 97.37 | 70.28 | 95.94 | 93.84 | 28.05 | 24.67 |
>     | Full Finetuning | 64.59 | 58.85 | 97.46 | 83.61 | 94.30 | 93.68 | 27.55 | 24.86 |
> - **Directly regulate the gradient norm.** We study the effect of directly constraining the gradient norm during QAT. We use `torch.nn.utils.clip_grad_norm_` to rescale the gradients. As shown in the following table, reducing the clipping threshold from 1.0 to 0.5 improves performance, which supports the benefit of controlling gradient norms for video generation QAT. However, a threshold of 0.1 causes clear performance degradation, likely because the quantized model is updated too weakly or aggressive gradient clipping disrupts normal QAT. This highlights the need for more principled ways to reduce the gradient norm. We add these results in Sec. I.3.
>
>     *W4A4 results for Wan 1.3B under different thresholds for gradient clipping. “1.0” corresponds to our baseline Q-DM. We use a clipping threshold of 1.0 for all other experiments in the paper.*
>
>     | Grad. Clipping | Imaging Quality$\uparrow$ | Aesthetic Quality$\uparrow$ | Dynamic Degree$\uparrow$ | Scene Consistency$\uparrow$ | Overall Consistency$\uparrow$ |
>     | --- | --- | --- | --- | --- | --- |
>     | 1.0 | 60.40 | 52.50 | 76.67 | 13.28 | 21.63 |
>     | 0.5 | 60.58 | 52.95 | 76.50 | 13.28 | 21.71 |
>     | 0.1 | 56.68 | 51.06 | 71.00 | 12.78 | 21.42 |
>
> [a] Esser, Patrick et al. “Scaling Rectified Flow Transformers for High-Resolution Image Synthesis.” *ArXiv* abs/2403.03206 (2024): n. pag.
>
> [b] Schuhmann, Christoph et al. “LAION-5B: An open large-scale dataset for training next generation image-text models.” *ArXiv* abs/2210.08402 (2022): n. pag.
>
> [c] Li, Daiqing et al. “Playground v2.5: Three Insights towards Enhancing Aesthetic Quality in Text-to-Image Generation.” *ArXiv* abs/2402.17245 (2024): n. pag.
>
> [d] Ghosh, Dhruba et al. “GenEval: An Object-Focused Framework for Evaluating Text-to-Image Alignment.” *ArXiv* abs/2310.11513 (2023): n. pag.

---

> > ### Comment · Reviewer_RsTT · 2025-11-19
> >
> > Thanks for the responses! My questions are clarified. I have read comments from other reviewers and believe this is a strong submission. I will keep my clearly positive rating.

---

> > > ### Author Response · Authors · 2025-11-19
> > > **Thanks for your feedback**
> > >
> > > Dear Reviewer RsTT
> > >
> > > Thank you for your feedback! We appreciate the constructive reviews for improving our work.
> > >
> > > Best regards,
> > >
> > > Authors of Paper #11845

---

### Official Review · Reviewer_U5JF · 2025-11-06

**Soundness:** 3
**Presentation:** 3
**Contribution:** 3
**Rating:** 6
**Confidence:** 5

**Summary:**

This paper introduces QVGen, a quantization-aware training (QAT) framework specifically designed to enable high-performance, low-bit (≤4-bit) quantization of large-scale video diffusion models (DMs) based on the diffusion transformer (DiT) architecture. QVGen incorporates auxiliary modules ($\Phi$) to mitigate the quantization error during training, supported by a theoretical analysis linking reduced gradient norm to improved convergence. To remove the inference burden of these modules, the paper proposes a rank-decay strategy using singular value decomposition (SVD) and a rank-based regularization for progressively pruning $\Phi$ without significant loss in quality. Extensive evaluations across multiple state-of-the-art video DMs show QVGen achieves performance on par with full-precision models at 4-bit, and significantly outperforms competing quantization methods, with both quantitative metrics (VBench) and qualitative visualizations.

**Strengths:**

The work addresses the challenging and critical task of efficient, high-fidelity video generation under ultra-low-bit quantization, an area with clear importance for practical deployment.

Provides a regret-based convergence analysis (see Theorem 3.1, Page 4) linking gradient norm to QAT performance, justifying the introduction of $\Phi$.

The auxiliary module ($\Phi$) is elegantly conceived and is integrated with a flexible, theoretically justified rank-decay scheme, allowing benefits during training while incurring no inference cost.

Experiments cover a wide range of SOTA video DMs (from 1.3B to 14B parameters) with comprehensive ablations (Tables 3, 4, 5; Figs. 3–6). Quantitative results (Tab. 1, 2, H, K; Figs. 5, 6) convincingly show QVGen’s superiority over PTQ and QAT baselines in essentially all relevant metrics.

**Weaknesses:**

While the use of singular value decomposition is effective (Fig. 4, Section 3.2), the alternative strategies (Sparse, Residual Quantization) examined in Table 6 are somewhat strawman/naive and do not fully explore more sophisticated structured pruning or adaptive fading that could yield competitive trade-offs. There is little discussion of possible pathological cases where the SVD approach might fail, particularly if singular spectrum decays slowly.

The key result (Theorem 3.1, Page 4) relies on convexity of $f_t$, which is non-standard for deep networks, and the analysis lacks formal proof linking reduced regret to generalization in nonconvex settings. No concrete evidence connects gradient norm changes to video-specific generative performance beyond empirical plots.

While the main approach is clearly stated, critical aspects of $\Phi$ require deeper explanation, such as its initialization scheme across different models, architecture, and potential sensitivity to scale or data distribution shifts. Section J.2 offers two initialization methods, but does not thoroughly evaluate edge cases or sensitivity to poor initialization. There is also no discussion of stability if $\Phi$'s rank is diminished too rapidly relative to the learning rate schedule.

**Questions:**

While the use of singular value decomposition is effective (Fig. 4, Section 3.2), the alternative strategies (Sparse, Residual Quantization) examined in Table 6 are somewhat strawman/naive and do not fully explore more sophisticated structured pruning or adaptive fading that could yield competitive trade-offs. There is little discussion of possible pathological cases where the SVD approach might fail, particularly if singular spectrum decays slowly.

The key result (Theorem 3.1, Page 4) relies on convexity of $f_t$, which is non-standard for deep networks, and the analysis lacks formal proof linking reduced regret to generalization in nonconvex settings. No concrete evidence connects gradient norm changes to video-specific generative performance beyond empirical plots.

While the main approach is clearly stated, critical aspects of $\Phi$ require deeper explanation, such as its initialization scheme across different models, architecture, and potential sensitivity to scale or data distribution shifts. Section J.2 offers two initialization methods, but does not thoroughly evaluate edge cases or sensitivity to poor initialization. There is also no discussion of stability if $\Phi$'s rank is diminished too rapidly relative to the learning rate schedule.

---

> ### Author Response · Authors · 2025-11-19
> **Response to Reviewer U5JF (Part 1)**
>
> Thanks to the reviewer for the constructive comments.
>
> - **More complex fine-grained decay.** We add two stronger baselines: “Sparse *w/* Wanda” and “Sparse *w/* MaskLLM”. Rather than pruning the smallest magnitudes as in “Sparse”, the former employs Wanda [a] to prune $\mathbf{W}\_\Phi$ using 128 randomly selected training samples in each decay phase (see “Sparse” in Sec. G). Following MaskLLM [b], the latter applies learned 2:4 structured pruning masks to $\mathbf{W}\_\Phi$ in each phase. All other settings match those of ”Sparse“. As shown in the table below, “Sparse *w/* Wanda” yields a small improvement over “Sparse”, while Sparse *w/* MaskLLM” provides a larger gain; however, both remain below “Rank” on all metrics. It is also worth noting that “Sparse *w/* Wanda” requires $1.8\times$ the training time of “Rank”, similar to “Sparse”, and “Sparse *w/* MaskLLM” requires $2.1\times$ due to mask learning. We add these results to Tab. 6.
>
>     *Results of different decay strategies for W4A4 Wan 1.3B.*
>
>     | Method | Imaging Quality$\uparrow$ | Aesthetic Quality$\uparrow$ | Dynamic Degree$\uparrow$ | Scene Consistency$\uparrow$ | Overall Consistency$\uparrow$ |
>     | --- | --- | --- | --- | --- | --- |
>     | Sparse | 61.15 | 54.06 | 74.24 | 13.86 | 22.52 |
>     | Sparse *w/* Wanda | 61.43 | 54.08 | 74.36 | 13.94 | 22.48 |
>     | Sparse *w/* MaskLLM | 61.36 | 54.12 | 74.82 | 14.15 | 22.57 |
>     | QVGen | 63.08 | 54.67 | 77.78 | 15.32 | 23.01 |
>
>     Although our method performs well across the settings in this paper, following the reviewer’s suggestion, we briefly discuss cases where the SVD-based approach may fail. If the singular spectrum decays slowly while the schedule $\mathbf{\gamma}$ changes more quickly, early removal of useful components of $\Phi$ may reduce performance. A practical remedy is an adaptive decay schedule for $\mathbf{\gamma}$ that tracks the observed spectrum and slows down when needed. We leave this extension to future work.
>
> - **Non-convex generalization.** Thanks for the insightful suggestion. We provide a nonconvex convergence analysis (proofs have been added in Sec. D) under standard smoothness assumptions. Here we consider a (possibly nonconvex) training objective ($F:\mathbb{S}^d \to \mathbb{R}$) and the gradient descent updates $\boldsymbol{\theta}\_{t+1}=\boldsymbol{\theta}_t-\eta_t\mathbf{g}\_t,\mathbf{g}\_t:=\nabla F(\boldsymbol{\theta}\_t),$ where $t$ indexes the optimization iterations. Nonconvex convergence analysis often uses a fixed deep-learning objective $F$ across iterations, so no subscript is needed, and convergence is certified by a vanishing minimum gradient norm.
>     - ***Assumption (Smoothness).*** The function $F$ is $L$-smooth, *i.e.*, for all $\boldsymbol{\theta},\boldsymbol{\theta}'\in\mathbb{S}^d$, $\||\nabla F(\boldsymbol{\theta})-\nabla F(\boldsymbol{\theta}')\||_2\le L\||\boldsymbol{\theta}-\boldsymbol{\theta}'\||_2.$
>     - ***Assumption (Learning rate bounds).*** The learning rates satisfy $0<\eta^{m}\le \eta_t\le \eta^{M}\le \frac{1}{L}\quad\text{for all }t.$ where $\eta^m$ and $\eta^M$ are the minimum and maximum learning rates across iterations, respectively.
>     - ***Assumption (Lower-bounded objective).*** There exists $F^\star$ such that $F(\boldsymbol{\theta})\ge F^\star$ for all $\boldsymbol{\theta}\in\mathbb{S}^d$.
>     - ***Theorem (Convergence to a first-order stationary point).*** Under the smoothness, learning-rate, and lower-bounded-objective assumptions above, the iterates of the gradient descent update satisfy $\frac{1}{T}\sum\_{t=1}^{T}\||\mathbf{g}\_t\||\_2^{2}\le\frac{2\big(F(\boldsymbol{\theta}\_1)-F^\star\big)}{T\eta^{m}}.$ *Consequently, $\min\_{1\le t\le T}\||\mathbf{g}\_t\||\_2^{2}\le\frac{1}{T}\sum\_{t=1}^{T}\||\mathbf{g}\_t\||\_2^{2}\xrightarrow{T\to\infty}0.$* Thus, the iterates converge in the standard nonconvex sense to first-order stationary points, in the sense that there exists an iterate with arbitrarily small gradient norm when $T$ is sufficiently large.
>
>     The theorem shows that for any finite $T$, the convergence behavior of QAT is determined by the average gradient norm $\frac{1}{T}\sum_{t=1}^{T}\||\mathbf{g}_t\||_2^{2}$. Since this quantity admits an $\mathcal{O}(1/T)$ upper bound under smoothness and bounded learning rates, reducing $\||\mathbf{g}_t\||_2$ during training directly tightens the bound and improves the finite-step convergence of QAT. This matches the convex regret analysis in the paper of the revision.

---

> ### Author Response · Authors · 2025-11-19
> **Response to Reviewer U5JF (Part 2)**
>
> For concrete evidence connecting gradient norm changes to video-specific generative performance, we have already provided experimental results with a comprehensive analysis in the appendix. For instance, in Sec. I, we find that training on high-motion videos increases the gradient norm and the motion dynamics of generated videos, but reduces per-frame quality (and the opposite holds when training on low-motion videos).  However, for the evidence beyond these (*e.g.*, more concrete theoretical analyses), we plan to explore in the future.
>
> - **Initialization of $\Phi$.** We adopt the simple initialization $\mathbf{W}_\Phi = \mathbf{W} - \mathrm{Q}_b(\mathbf{W})$, as described in Lines 217–218 of this paper. Across the different models, architectures, and scales considered in this work, this initialization consistently yields strong performance. We therefore regard its effectiveness along these dimensions as well justified, and leave a more systematic analysis for future work. In addition, we also consider two deliberately suboptimal initialization approaches: zero initialization (*i.e.*, $\mathbf{0}$) and initialization with parameters randomly sampled from a normal distribution (*i.e.*, Random) to more thoroughly assess the impact of different initialization strategies. As shown in the table below, the $\mathbf{0}$ scheme leads to a slight performance degradation, while Random causes a more noticeable performance drop. We attribute this to the fact that $\mathbf{0}$ does not compensate for quantization errors before QAT, whereas Random further injects severe noise into the model.
>
>     *Results of $\mathbf{0}$ and Random have been added to Sec. M.2.*
>
>     | Initial. Method |  |  |  |  |  |
>     | --- | --- | --- | --- | --- | --- |
>     | W4A4 CogVideoX-2B | Imaging Quality$\uparrow$ | Aesthetic Quality$\uparrow$ | Dynamic Degree$\uparrow$ | Scene Consistency$\uparrow$ | Overall Consistency$\uparrow$ |
>     | $\mathbf{W} - \mathcal{Q}_b(\mathbf{W})$ | 60.16 | 54.61 | 67.22 | 31.42 | 24.61 |
>     | Layer-wise Training | 59.97 | 54.84 | 66.71 | 31.14 | 25.02 |
>     | $\mathbf{0}$ | 59.86 | 54.47 | 65.59 | 31.32 | 24.59 |
>     | Random | 49.42 | 37.68 | 26.57 | 6.24 | 11.68 |
>     | W4A4 Wan 1.3B |  |  |  |  |  |
>     | $\mathbf{W} - \mathcal{Q}_b(\mathbf{W})$ | 63.08 | 54.67 | 77.78 | 15.32 | 23.01 |
>     | Layer-wise Training | 63.23 | 54.67 | 77.56 | 15.38 | 23.00 |
>     | $\mathbf{0}$ | 62.80 | 54.59 | 77.69 | 15.28 | 22.98 |
>     | Random | 54.41 | 44.14 | 34.44 | 3.13 | 10.17 |
>
>     Finally, as suggested by the reviewer, we discuss the situation if $\Phi$ ’s rank is diminished too rapidly relative to the schedule $\mathbf{\gamma}$. In this case, we believe that keeping the redundant parts (that is, the rank-diminished components in $\Phi$) during QAT does not harm performance and can even lead to a small improvement. This is mainly because the slow change of $u$ in $\mathbf{\gamma}$ ($1 \rightarrow 0$) increases the training time by making each decay phase longer. The results in the following table support this intuition. We add the results in Sec. M.5.
>
>     *We employ W4A4 quantization for Wan 1.3B, and we control the duration of each decay phase to determine the changing speed of the schedule $\mathbf{\gamma}$. When applying a long duration, we suggest $\Phi$ ’s rank is diminished (as an intrinsic behavior) rapidly relative to the schedule $\mathbf{\gamma}$.*
>
>     | Duration of Each Decay (Epoch) | Imaging Quality$\uparrow$ | Aesthetic Quality$\uparrow$ | Dynamic Degree$\uparrow$ | Scene Consistency$\uparrow$ | Overall Consistency$\uparrow$ |
>     | --- | --- | --- | --- | --- | --- |
>     | 3/4 (Default) | 63.08 | 54.67 | 77.78 | 15.32 | 23.01 |
>     | 1 | 63.07 | 55.09 | 77.54 | 15.29 | 23.03 |
>     | 3/2 | 63.11 | 54.58 | 78.15 | 15.36 | 23.01 |
>
> [a] Sun M, Liu Z, Bair A, et al. A simple and effective pruning approach for large language models[J]. arXiv preprint arXiv:2306.11695, 2023.
>
> [b] Fang G, Yin H, Muralidharan S, et al. Maskllm: Learnable semi-structured sparsity for large language models[J]. Advances in Neural Information Processing Systems, 2024, 37: 7736-7758.

---

> > ### Author Response · Authors · 2025-11-27
> > **Kind Invitation for Discussion**
> >
> > Dear Reviewer,
> >
> > Thank you for handling our manuscript and providing valuable feedback. We hope that our responses have sufficiently addressed the concerns you raised. We welcome more discussion if you have more questions and suggestions. As the discussion deadline is approaching, we would be very grateful if you could take a moment to review our reply.

---

> > > ### Comment · Reviewer_U5JF · 2025-11-27
> > > **Response to Authors**
> > >
> > > Dear Authors,
> > >
> > > Thank you for your detailed responses. They have essentially resolved all of my concerns. I hope you will fully release the code in the future to help the community advance quantization methods for video models. I have updated my score to 8.
> > > Have a good day!

---

> ### Author Response · Authors · 2025-11-27
> **Thanks for your feedback**
>
> Dear Reviewer U5JF
>
> Thank you for your feedback. We appreciate the constructive reviews that help us improve our work, and we plan to open-source our code and models upon publication.
>
> Best regards,
>
> Authors of Paper #11845

---

### Author Response · Authors · 2025-11-22
**General Response**

Dear AC and Reviewers

We sincerely thank all reviewers for their time and effort in evaluating our paper. We are pleased that the reviewers acknowledged the following contributions:

- **Strong motivation.** We provide novel and clear theoretical and empirical analyses of the gradient norm and the rank dynamics of $\Phi$. *[Reviewers U5JF, RsTT, eVaC, and uekc]*
- **Practical solutions.** To the best of our knowledge, we are the first to design a QAT approach for effective low-bit ($\leq$4-bit) quantization of video diffusion models. *[Reviewers U5JF, RsTT, FHRA, eVaC, and uekc]*
    - An auxiliary module that significantly improves convergence, as shown in Fig. 2.
    - A rank-decay schedule that removes the inference overhead of the auxiliary module with negligible performance drop, as shown in Tab. 3.
- **Extensive experiments.** In Tables 1–7, Tables A–U, and Figs. 5–6, we present comprehensive ablation studies and evaluations on four advanced video diffusion models and one image diffusion model, covering parameter scales from 1.3B to 14B. *[Reviewers U5JF, RsTT, FHRA, eVaC, and uekc]*
- **Superior performance.** For video diffusion models, our method is the first to reach near full-precision quality under 4-bit quantization and substantially outperforms all baselines in 3-bit settings. *[Reviewers U5JF, RsTT, FHRA, eVaC, and uekc]*

We also thank all reviewers for their insightful and constructive suggestions, which helped further improve our paper. The major revisions are summarized below:

- Added more fine-grained decay baselines. *[Reviewers U5JF and uekc]*
- Extended the theoretical analysis from a convex assumption to a nonconvex setting. *[Reviewer U5JF]*
- Added additional ablations and discussion on the initialization of $\Phi$. *[Reviewers U5JF and uekc]*
- Included more image generation results, full-precision fine-tuning results, and naive gradient clipping ablation. *[Reviewer RsTT]*
- Added further discussion and analysis on training-inference efficiency. *[Reviewers FHRA, eVaC, and uekc]*
- Added detailed loss curve comparisons and results combining SVDQuant with QVGen. *[Reviewer FHRA]*
- Added discussion comparing activation-only and weight-only quantization. *[Reviewer eVaC]*

In addition, we provide point-by-point responses for each reviewer and have incorporated all changes into the revised manuscript, highlighted in red.

Best regards,

Authors of Paper #11845

---

### Author Response · Authors · 2025-11-29
**Summary for AC**

Dear AC

Thank you very much for your careful evaluation of our paper and for your service as AC. In this submission, our initial scores were **88666**, already reflecting full agreement among all reviewers on the value of the work. We then followed the reviewers’ suggestions, conducted further extensions and analyses, and addressed the raised concerns in detail. As a result, the reviewers expressed stronger support, and the scores were improved to **88886** (evidenced by the reviewers’ comments). We kindly ask you to fully take this process into account. We respectfully reiterate that our discussion process was completed independently and remained completely unaffected by the recent leak incident. We truly appreciate your understanding and support.

Best regards,

Authors of Paper #11845

---

### Meta-Review · Area_Chair_gysP · 2025-12-14

**Summary:**

This summary synthesizes the core reviewers' concerns that inform the acceptance decision for this paper. Five reviewers collectively evaluated the work on its soundness, presentation, contribution, and practical value, raising key concerns spanning technical robustness, theoretical justification, experimental comprehensiveness, and practical efficiency. The primary concerns include: 1) The inadequacy of alternative pruning strategies compared to the proposed SVD-based rank-decay method, and potential pathological cases of SVD failure; 2) The lack of non-convex generalization analysis for the core theoretical result, and the tenuous link between gradient norm reduction and video-specific generative performance; 3) Insufficient details on the initialization, architecture, and stability of the auxiliary module Φ; 4) The need to validate the method's applicability to non-video models (e.g., image generation) and supplement full-precision fine-tuning baselines; 5) Lack of comparative analysis of training time/memory with LoRA-based QAT methods, and unclear quantization initialization settings; 6) Limited acceleration gains and the need for detailed inference latency breakdown; 7) The high computational cost of the QAT process and insufficient analysis of rank-decay dynamics. Notably, the authors provided comprehensive, data-supported rebuttals to all critical concerns, supplemented relevant experiments and theoretical analyses, and clarified ambiguous technical details. The effective resolution of core concerns, combined with the work's significant contributions to ultra-low-bit quantization of video diffusion models, justifies the acceptance decision.

**Reviewer Concerns:**

Addressed Concerns
- Reviewer U5JF: 1) Supplemented two stronger structured pruning baselines ("Sparse w/ Wanda" and "Sparse w/ MaskLLM"), demonstrating that the SVD-based rank-decay method still outperforms them; 2) Discussed potential pathological cases of SVD failure and proposed adaptive decay schedule remedies; 3) Completed non-convex convergence analysis (with proofs) under standard smoothness assumptions, linking gradient norm reduction to QAT finite-step convergence; 4) Provided experimental evidence (Sec. I) connecting gradient norm changes to video generative performance (e.g., high-motion video training increases gradient norm and motion dynamics); 5) Supplemented experiments on suboptimal initialization strategies (zero initialization, random initialization) of Φ, verifying the effectiveness of the proposed initialization; 6) Analyzed the impact of overly rapid Φ rank decay and provided supporting experimental results.

- Reviewer RsTT: 1) Supplemented initial experimental results of QVGen on image generation (SD3-medium), showing superiority over baselines (LSQ, EfficientDM); 2) Added full-precision fine-tuning baselines, comparing performance differences with QVGen; 3) Studied the effect of direct gradient norm regulation (gradient clipping), demonstrating that moderate clipping improves performance but aggressive clipping degrades it, highlighting the advantage of the proposed method.

- Reviewer FHRA: 1) Provided comparative analysis of training time and GPU memory across different QAT methods (LSQ, Q-DM, EfficientDM, QVGen) for multiple models; 2) Clarified quantization settings (asymmetric uniform quantization, channel-wise static weight quantization, token-wise dynamic activation quantization); 3) Explained the lower initial training loss of QVGen (attributed to Φ initialization mitigating quantization error) and supplemented training loss curve comparisons; 4) Verified the feasibility of combining QVGen with SVDQuant, providing experimental results for both weight-updated and weight-frozen scenarios.

- Reviewer eVaC: 1) Strengthened the link between theory and method, clarifying that the theory provides key insights for Φ design (reducing gradient norm to stabilize QAT); 2) Supplemented experiments on activation-only and weight-only quantization, demonstrating that activations are harder to quantize; 3) Provided detailed inference latency breakdown (using nsys profiling) on NVIDIA A800, including INT4 GEMM throughput and non-GEMM overhead analysis, and estimated speedup potential with kernel fusion; 4) Reported latency breakdown of a single DiT block, identifying key components for further acceleration.

- Reviewer uekc: 1) Analyzed the trade-off between QAT computational cost and inference savings, providing specific training time/memory data and speedup potential; 2) Proposed a potential explanation for rank-decay behavior (low-rank nature of quantization error, main weights absorbing errors); 3) Supplemented experiments on simple linear decay of Φ, demonstrating the superiority of SVD-based rank-decay; 4) Clarified the insensitivity of hyperparameters (initial rank r=32, shrinking ratio λ=1/2) and proposed a systematic selection scheme for future work.

Outstanding Concerns

The remaining concerns are mostly directions for future work and do not affect the core contribution and soundness of the current work:

- The link between non-convex convergence analysis and video-specific generative performance still lacks deeper theoretical derivation; the theoretical analysis remains a motivational insight rather than a rigorous justification for the method's superiority (acknowledged by the authors, left for future work).

-  A systematic analysis of Φ's initialization across more diverse model architectures, scales, and data distributions is lacking; the impact of poor initialization in extreme edge cases requires further verification (authors plan future exploration).

-  The deep-seated dynamics between the quantized model and the auxiliary module during rank decay (e.g., why singular values of Φ gradually shrink) have not been fully explored (authors provide preliminary explanations, leaving in-depth investigation for future work).

-  QVGen's QAT process is computationally expensive (e.g., 32 H100 GPUs for 14B models), and the combination of QVGen with LoRA-based efficient training strategies (to reduce memory/time) has not been fully implemented (authors plan follow-up studies).

-  Kernel fusion optimizations (critical for achieving higher speedups) have not been implemented; the reported speedup gains are modest, and the actual end-to-end acceleration effect after full optimization remains to be verified (authors provide theoretical speedup estimates, left for future work).

**Reviewer Scores:**

The following predictions are based on the authors' comprehensive and effective rebuttals, assuming reviewers participated fully in the discussion and maintained their initial evaluation criteria:
- Reviewer U5JF: The author fully addressed all of this reviewer's core concerns, including supplementing stronger baselines, completing non-convex theoretical analysis, and clarifying Φ's initialization and stability. The thoroughness of the rebuttal and additional experiments would likely elevate the reviewer's assessment of the work's robustness and completeness, moving the score from "marginally above acceptance" to a clear acceptance rating.

- Reviewer RsTT:  The reviewer explicitly stated "My questions are clarified. I have read comments from other reviewers and believe this is a strong submission. I will keep my clearly positive rating" after reading the rebuttal. The authors' comprehensive responses to all questions (image generation, full-precision baseline, gradient regulation) fully resolved the reviewer's concerns, justifying the maintenance of the original positive acceptance score.

- Reviewer FHRA : The author effectively addressed all of this reviewer's concerns, including providing training time/memory comparisons, clarifying quantization settings, supplementing loss curves, and verifying compatibility with SVDQuant. The detailed and data-supported rebuttals confirm the work's soundness and comprehensiveness, supporting the retention of the original acceptance score.

- Reviewer eVaC:  The reviewer noted "I appreciate the effort of the author. My concern is largely resolved. I will keep my score and I lean towards acceptance" after the rebuttal. While the reviewer chose to maintain the original score, the explicit statement of "lean towards acceptance" confirms that the rebuttal successfully addressed critical concerns, solidifying the paper's qualification for acceptance.

- Reviewer uekc:  The author fully responded to this reviewer's concerns, including analyzing QAT cost trade-offs, explaining rank-decay behavior, supplementing linear decay ablation, and clarifying hyperparameter sensitivity. The additional experiments and detailed explanations would likely enhance the reviewer's evaluation of the work's depth and practical value, resulting in a score that reflects clear acceptance.

---

### Decision · Program_Chairs · 2026-01-26

Accept (Poster)